# EMMA: EMPOWERING MULTI-MODAL MAMBA WITH STRUCTURAL AND HIERARCHICAL ALIGNMENT

**Yifei Xing**[1,2,3], **Xiangyuan Lan**[2,5,*], **Ruiping Wang**[1,3,*], **Dongmei Jiang**[2], **Wenjun Huang**[4]
**Qingfang Zheng**[2], **Yaowei Wang**[2]
[1]Institute of Computing Technology, Chinese Academy of Sciences, [2]Pengcheng Laboratory,
[3]University of Chinese Academy of Sciences, [4]Sun Yat-sen University [5]Pazhou Laboratory (Huangpu)
`xingyifei22@mails.ucas.ac.cn`,
`{lanxy, jiangdm, zhengqf01, wangyw}@pcl.ac.cn`
`wangruiping@ict.ac.cn, huangwj98@mail2.sysu.edu.cn`

## ABSTRACT

Mamba-based architectures have shown to be a promising new direction for deep learning models owing to their competitive performance and sub-quadratic deployment speed. However, current Mamba multi-modal large language models (MLLM) are insufficient in extracting visual features, leading to imbalanced cross-modal alignment between visual and textural latents, negatively impacting performance on multi-modal tasks. In this work, we propose **E**mpowering **M**ulti-modal **M**amba with Structural and Hierarchical **A**lignment (EMMA), which enables the MLLM to extract fine-grained visual information. Specifically, we propose a pixel-wise alignment module to autoregressively optimize the learning and processing of spatial image-level features along with textual tokens, enabling structural alignment at the image level. In addition, to prevent the degradation of visual information during the cross-model alignment process, we propose a multi-scale feature fusion (MFF) module to combine multi-scale visual features from intermediate layers, enabling hierarchical alignment at the feature level. Extensive experiments are conducted across a variety of multi-modal benchmarks. Our model shows lower latency than other Mamba-based MLLMs and is nearly four times faster than transformer-based MLLMs of similar scale during inference. Due to better cross-modal alignment, our model exhibits lower degrees of hallucination and enhanced sensitivity to visual details, which manifests in superior performance across diverse multi-modal benchmarks. Code provided at https://github.com/xingyifei2016/EMMA.

## 1 INTRODUCTION

Recently there has been a notable increase in the development of domain-general AI agents Kalla et al. (2023); Zhao et al. (2023a) which can simultaneously solve a diverse range of tasks and exhibit superior performance. Among them, multi-modal large language models (MLLMs) Achiam et al. (2023); Team et al. (2023); Liu et al. (2024b) have emerged as a promising direction due to their effectiveness in visual perception and logical reasoning. MLLMs usually consist of an image encoder that converts images to visual tokens and a strong large language model (LLM) backbone to process the visual and textual tokens concurrently. This integration of visual and textual information not only enhances the understanding of visual content but also provides a more comprehensive context for language understanding and generation. As a result, these cross-modal models have consistently achieved state-of-the-art performances in tasks such as image captioning Hossain et al. (2019), visual reasoning Johnson et al. (2017), and visual question answering Antol et al. (2015).

However, a gravid challenge for current MLLMs is the substantial computational cost associated with training and deployment Achiam et al. (2023); Han et al. (2022). In fact, current MLLMs are predominately transformer-based, and consequently suffer from an input-dependent attention mechanism that is quadratic in complexity Katharopoulos et al. (2020b). Transformers also struggle with

---
*Corresponding authors

capturing long-ranged dependencies in data due to a limited context window Zimerman & Wolf (2023); Tay et al. (2020). Many works have attempted to address these issues and propose novel architectures to increase effectiveness in long-sequence processing while reducing both computational and memory costs Wang et al. (2020); Shen et al. (2021b); Li et al. (2020); Gu & Dao (2023).

Mamba Gu & Dao (2023); Dao & Gu (2024) is a structured state-space sequence model (SSM) proposed as an alternative for transformers Aoki (2013); Gu et al. (2021a;b) and has demonstrated superior performance in sequential tasks such as language and genomics Ma et al. (2024b); Gu & Dao (2023). Inspired by classical state-space models Kalman (1960), Mamba is efficiently formulated as either a recurrence or convolution with near-linear scaling in sequence length, while also possessing principled mechanisms for modeling long-ranged dependencies. Furthermore, Dao & Gu (2024); Ali et al. (2024) have addressed the close connection between Mamba and variants of attention mechanisms in transformers, suggesting a promising new underlying architecture for previously transformer-dominated models that is more efficient and achieves competitive performances.

However, while Mamba models possess superior performance on sequential textual data, they remain ineffective in processing visual data, especially in large and huge models Ren et al. (2024); Zhu et al. (2024a); Liu et al. (2024c) where performance degrades quickly when in the billion-parameter scale. This causes an imbalance in the quality of visual and texture latents in the Mamba LLM, where coarse visual features are ineffectively aligned with better-quality textual features. Furthermore, while transformer-based LLM models contain positional embeddings to guide the learning of spatial information, Mamba LLM layers lack structural constraints on the visual features, causing deeper layers of the LLM to gradually lose fine-grained spatial integrity in visual features, as shown in Fig. 1. This eventually leads to an increasingly blurred feature during the final layers of the LLM, which weakens the cross-modal alignment between visual and textual cues, ultimately resulting in suboptimal performance and higher degrees of hallucination (Fig. 1). With respect to improving the quality of visual latents, current Mamba-based MLLMs employ various strategies such as incorporating multiple vision encoders Zhao et al. (2024a) or integrating a visual selective scan within the projection MLP Qiao et al. (2024) to impose structural constraints. However, these approaches primarily focus on preserving the visual feature *prior* to it's integration into the Mamba LLM, lacking mechanisms to maintain its quality *within* the MLLM, where the visual latents participates in the autoregressive generation of textual tokens.

Building on these observations, we propose **E**mpowering **M**ulti-modal **M**amba with Structural and Hierarchical **A**lignment (EMMA) that addresses the insufficient extraction of visual information *inside* the Mamba LLM. First, we note the autoregressive nature of Mamba models that allows for an analogous structure for next-token prediction in the visual domain, which has been shown to be effective in vision-related tasks Ren et al. (2024); El-Nouby et al. (2024). Thus, we extend the prediction of text tokens to the prediction of the visual image as well, to serve as a **structural** constraint on the visual latents for the Mamba LLM. A pixel-wise alignment is proposed to enforce the preservation of key spatial and structural information during training. Next, to further mitigate the gradual loss of fine-grained visual features in intermediate LLM layers, we propose a Multiscale Feature Fusion (MFF) module that **hierarchically** preserves fine-grained visual features. By building additional contribution of intermediate layers in the final pixel-wise alignment loss, the MFF effectively alleviates gradual information loss, resulting in finer-grained visual features. The major contributions of this paper are three-fold:

- We observe the imbalance between the quality of image and text latents within MLLM Mamba models and propose a pixel-wise alignment to autoregressively encourage the learning and processing of structural visual features, ultimately enabling better cross-modal alignment of visual and textural latents.

- To mitigate the gradual loss of fine-grained visual cues in the LLM, we propose a Mamba-based multi-scale fusion module that combines visual features from multiple intermediate layers, resulting in a richer representation for the final alignment phase.

- We conduct extensive experiments on various multi-modal benchmarks, comparing against current state-of-the-art multi-modal LLMs. Due to better cross-modal alignment, EMMA achieves competitive performance to similar-scaled transformers and outperforms other Mamba models on a variety of tasks while achieving high inference speed.

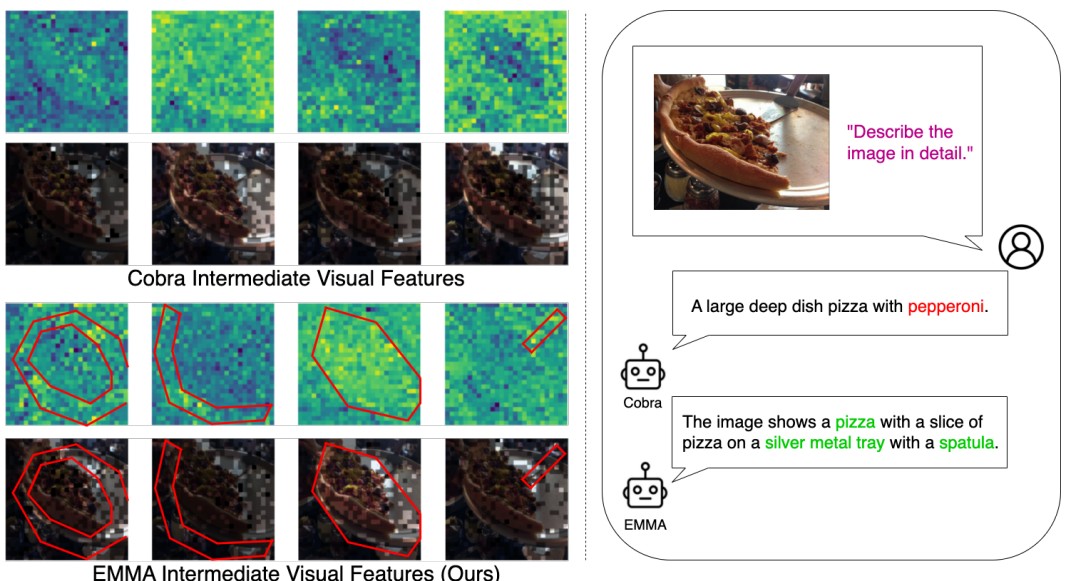

Figure 1: Given an image of a pizza and the prompt "Describe the image in detail", we visualize intermediate visual features and their corresponding textual responses in Mamba-based MLLMs. Each upper row represents the magnitude of reconstructed spatial activations on each image, and each bottom highlights these patches on the original image. Each column (from left to right) represents intermediate layers with increasing depth. **Cobra** Zhao et al. (2024a) experiences a gradual loss of visual features as visual cues become blurred and unrecognizable, resulting in ineffective cross-modal alignment and producing an overly generalized (and hallucinated) answer. On the other hand, due to better cross-modal alignment, **EMMA** is capable of preserving visual details even in deeper layers of the LLM, highlighting areas such as the perimeter of the pizza tray, the overall pizza, and the spatula on the top right of the image. The resulting text demonstrates higher alignment to the image data in the form of sensitivity to visual details and spatial relationships.

## 2 RELATED WORKS

### 2.1 MAMBA-BASED MODELS

Mamba Dao & Gu (2024); Gu & Dao (2023) is a form of state-space model that originally stemmed from classical signal processing theory Kalman (1960). These sequential models inherently excel at capturing long-range dependencies and are designed to be computationally efficient Gu et al. (2020; 2021b;a); Gupta et al. (2022). Expanding upon these works, Mamba Gu & Dao (2023) introduces a more effective selection mechanism that is input-dependent and a hardware-aware framework that is computationally efficient. Since then, the Mamba architecture has been successfully applied towards a variety of sequential tasks including NLP Yuan et al. (2024); Behrouz et al. (2024), speech Zhang et al. (2024a); Chen et al. (2024c), and motion Wang et al. (2024); Zhang et al. (2024b), rivaling the performance of transformer-based models while being more computationally efficient Qu et al. (2024). Mamba has also been adapted to non-sequential data such as vision and point clouds Xu et al. (2024a); Yang et al. (2024); Xing et al. (2024); Han et al. (2024), which differs by not adhering to any particular ordering Huang & Schneider (2011). These tasks are more challenging due to the lack of suitable and appropriate structural constraints in Mamba layers to preserve order-invariant information, such as position encoding in their transformer counterparts Vaswani (2017). Visual State Space model Liu et al. (2024c) introduces a novel 2D Selective Scan (SS2D) module that gathers contextual visual information from various perspectives. Vision Mamba Zhu et al. (2024a) applies Mamba to the Vision Transformer architecture and proposes a bi-directional SSM to better process visual features. However, despite gaining success in small-scale scenarios with less than 100M parameters, these vision Mamba models often falter when scaled-up in more complex scenarios. Ren et al. (2024) enhances Mamba's visual capability through autoregressive pretraining, which results in higher classification accuracy over its supervised-trained counterparts and successfully scales vi-

sual Mamba models to large and even huge model sizes. In this work, we explore ways to apply the Mamba architecture to multi-modal learning tasks, where large-scale processing of visual features in Mamba layers is required for aligning visual and language modalities.

## 2.2 MULTI-MODAL LARGE LANGUAGE MODELS (MLLMs)

MLLMs take advantage of a powerful large language model (LLM) Chang et al. (2024); Achiam et al. (2023) to simultaneously process data from both visual and textual modalities. Cho et al. (2021); Wang et al. (2022) initially proposed a unified transformer architecture for joint visual and language reasoning tasks that demonstrated better generalizability than conventional vision models. Since then, studies on MLLMs have established strong baselines for general-AI assistants Liu et al. (2024b); Achiam et al. (2023); Alayrac et al. (2022), which typically consist of a vision encoder, a cross-modal projector, and an LLM backbone. While most current MLLMs utilize a text-only loss function for optimization Li et al. (2024b); Zhu et al. (2024b); Lin et al. (2024), EMU Sun et al. (2023a; 2024) takes interleaved visual and textual inputs in a unified training object of predicting next text or image token in an autoregressive way. However, they require multiple training stages and additional computational cost for training a Stable Diffusion Rombach et al. (2022) visual decoder.

A major drawback of current MLLMs resides in the inherent computational overhead Katharopoulos et al. (2020b), predominantly attributed to their foundation in transformer architectures. Ma et al. (2024a); Ning et al. (2024) propose architectural changes to enable more efficient processing of data in MLLMs. Transformer-based MLLMs also struggle with learning long-range dependencies Zhu et al. (2021); Gu & Dao (2023), which is crucial for vision-language tasks. Chen et al. (2024b) repurpose the image embedding projector to encode long textual context, improving the MLLM's ability to handle longer sequences. Nevertheless, current progress in improving the computational complexity and learning long-range dependencies mostly depends on architectural changes in the transformer backbone. In this work, we explore how to effectively integrate the Mamba model into the transformer-dominated MLLM domain. Mamba offers a three to five times higher throughput than its transformer counterparts, while capable of processing million-length sequential data.

## 2.3 MAMBA-BASED MLLMs

Mamba has been extended to MLLMs due to their capabilities in both vision and NLP. Current multi-modal Mamba models Qiao et al. (2024); Zhao et al. (2024a) follow a simple LLaVA-based regimen Liu et al. (2024b) through a pretrained image encoder, Mamba LLM backbone, and a llama-like textual loss. These models utilize efficient down-sampling techniques Chu et al. (2024); Zhao et al. (2024a) and multi-direction scanning mechanisms Liu et al. (2024c); Qiao et al. (2024) as cross-modal projection modules to align visual and textual information. However, they lack explicit visual supervision during training. Given that Mamba models are inherently less effective in processing visual information in large-scale settings Ren et al. (2024), this results in poor-quality visual feature that weakens cross-modal alignment in Mamba MLLMs that easily surpass billions of parameters. Our work seeks ways to address the current bottleneck in mamba-based MLLMs where the mamba LLM extracts insufficient visual feature details that results in disparity in the alignment between images and text features. To this end, we impose additional structural constraints on visual features generated by the Mamba LLM, effectively enhancing the overall efficacy of multi-modal representations and cross-modal alignment.

## 3 METHOD

In this section, we present the overall framework for EMMA. EMMA aims to empower the insufficient extraction of visual information in Mamba MLLMs through **structural** and **hierarchical** alignment. We first provide the model architecture in Sec. 3.1. Then, we describe our pixel-level visual alignment for preserving structural visual features in Sec. 3.2, and multi-scale fusion module that hierarchically constrains intermediate features for retaining fine-grained visual cues in Sec. 3.3.

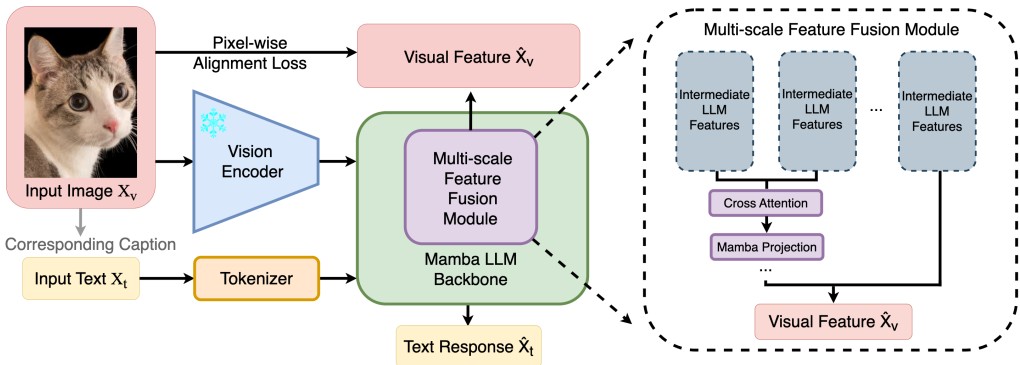

Figure 2: Overview of EMMA. In addition to the textual responses, our model extracts a holistic visual feature from the Mamba LLM through a multi-scale feature fusion module that hierarchically combines intermediate visual features through cross-attention and Mamba projections. A pixel-wise alignment loss is calculated between the final visual feature and the original image to enable the learning of more fine-grained features and increased multi-modal alignment.

## 3.1 ARCHITECTURE

Inspired by Zhao et al. (2024a); Qiao et al. (2024); Liu et al. (2024b), the architecture of EMMA consists of a pre-trained vision encoder $f_v$, a projection MLP $M_{proj}$, a pre-trained Mamba LLM $f_\phi$ parameterized by $\phi$, and a multi-scale fusion module $\psi$, as shown in Fig. 2. Taking an image $X_v$ and its corresponding tokenized caption $X_t$ as input, we first obtain visual features through the visual encoder:

$$\tilde{X}_v = f_v(X_v) \tag{1}$$

The MLP then maps visual tokens into the word embedding space, which is concatenated with the tokenized captions to form a multi-modal input. The combined tokens are fed into the Mamba LLM:

$$\tilde{X}_{LLM} = concat(\tilde{X}_v, X_t), \quad \hat{X} = f_\phi(\tilde{X}_{LLM}, \psi) \tag{2}$$

where the multimodal response $\hat{X}$ can be separated into a textual response $\hat{X}_t$ and a visual feature $\hat{X}_v$. Note that the visual feature is generated through the multi-scale fusion module which combines intermediate LLM features to alleviate the gradual loss of fine-grained visual features. Lastly, a pixel-wise alignment loss is applied to the visual feature $\hat{X}_v$ for preserving structural visual cues, and an autoregressive NLP loss is applied to the textural feature $\hat{X}_t$. More implementation details can be found in Appendix.

## 3.2 STRUCTURAL ALIGNMENT VIA PIXEL-WISE ALIGNMENT LOSS

We first consider the standard Mamba MLLM training procedure, given multimodal input $\tilde{X}_{LLM}$ and Mamba LLM $f_\phi$:

$$f_\phi(\tilde{X}_{LLM}) = [\hat{X}_v, \hat{X}_t] = \hat{X} \tag{3}$$

The target text token sequence $\hat{X}_t = \{\hat{x}_{t,i}\}_i^L$ of length $L$ is generated by computing the following probability and optimized by minimizing the corresponding negative log-likelihood function $\mathcal{L}_{text}$:

$$p\left(\hat{X}_t \mid X_v, X_t\right) = \prod_{i=1}^{L} p_\phi\left(\hat{x}_{t,i} \mid X_v, \{X_{t,j} \mid j < i\}\right), \quad \mathcal{L}_{text} = -\log p\left(\hat{X}_t \mid X_v, X_t\right) \tag{4}$$

where $\hat{x}_{t,i}$ depends on the visual features $X_v$ and the set of previous text tokens $\{X_{t,j} \mid j < i\}$. We observe the lack of supervision on image-level features, as they serve solely as conditional prior for predicting text tokens $\hat{X}_t$. As Mamba models inherently struggle with visual tasks in larger models Ren et al. (2024), this may result in poor quality image features during cross-modal alignment, negatively impacting model performance. It becomes crucial to construct additional constraints on the image-level features for better cross-modal alignment. Our insight is to condition the generation

of visual features *structurally*, which concern the organization, arrangement, or spatial relationships present within visual data. To achieve this, we extend the paradigm of next-token prediction to enable the joint generation of textural tokens and visual images. Thus, similar formulation arises for the generation of a target visual sequence $\hat{X}_v = \{\hat{x}_{v,i}\}_i^K$ of length $K$, given by the following probability function:

$$p\left(\hat{X}_v \mid X_v, X_t\right) = \prod_{i=1}^{K} p_\phi\left(\hat{x}_{v,i} \mid \{X_{v,j} \mid j < i\}, X_t\right) \tag{5}$$

We then project the generated visual feature $\hat{X}_v$ from word embedding space back to the image domain through a Mamba-based decoder $f_{dec}$ to retrieve the final generated visual sequence. Using the original image as reference, a common loss function for measuring the quality and consistency between two images is $L2$ distance. The $L2$ loss considers global features of the image, focusing on overall similarity that helps in preserving structures and shapes in the image. We formulate our **Pixel-wise Alignment Loss** $\mathcal{L}_{pixel}$ to constrain the decoded image to match the original image as:

$$\mathcal{L}_{pixel} = \|f_{dec}(\hat{X}_v) - X_v\|_2^2 \tag{6}$$

By imposing our pixel-wise alignment loss, we constrain the output visual feature to match closely with the original image, forcing the model to understand and preserve structural features of the visual input, and effectively increasing the quality of visual latents for cross-modal alignment. Unlike Zhan et al. (2024); Sun et al. (2023a), our visual decoder is extremely small-scaled and efficient, and does not require a separate stage for training.

### 3.3    Hierarchical Alignment via Multi-scale fusion module

While the pixel-wise alignment loss forces the Mamba LLM to retain structural characteristics of the visual input, we find that Mamba LLMs have a tendency to gradually lose fine-grained visual details through intermediate layers, as shown in Fig. 1. This could be due to the inherent lack of 'positional embedding'-analogues within the Mamba LLM backbone, such that spatial and fine-grained information is more easily distorted inside the LLM. Consequently, we contend that the visual feature derived from the final layer of Mamba LLM alone proves inadequate in preserving fine visual intricacies. To enable the Mamba LLM to extract more sufficient visual details and prevent gradual feature loss from the image modality, we devise a **Multi-scale Feature Fusion** (MFF) module that *hierarchically* integrates multiple intermediate features of the pretrained visual encoder for the visual alignment. Combining multi-level features that concern different levels of granularity enables the model to effectively capture intricate details at various scales, enhancing its ability to comprehend and interpret complex visual information with greater precision. Specifically, our multi-scale feature fusion module consists of I fusion blocks $\psi = \{\mathcal{B}_i\}^I$, where each $\mathcal{B}$ can be regarded as a tiny Mamba network consisting of a cross-attention module and a Mamba layer, which combines intermediate features in a pairwise fashion. For instance, given hidden visual features $\{\overline{X}_i, \overline{X}_j, \overline{X}_k\}$ from layers $i$, $j$, $k$ and the corresponding two-layer MFF $\psi = \{\mathcal{B}_i\}^2$, the final aggregated output of the features $\overline{X}_v$ is generated as follows:

$$\overline{X}_v = \psi(\overline{X}_i, \overline{X}_j, \overline{X}_k) = \mathcal{B}_2(\mathcal{B}_1(\overline{X}_i, \overline{X}_j), \overline{X}_k) \tag{7}$$

where each block $\mathcal{B}$ is given by residually-connected Mamba and cross attention layers:

$$\mathcal{B}(X,Y) = \hat{\mathcal{B}}(X,Y) + Mamba(\hat{\mathcal{B}}(X,Y)); \quad \hat{\mathcal{B}}(X,Y) = X + cross\_attn(X,Y) \tag{8}$$

In practice, we utilize three intermediate layers along with the final output layer for producing the aggregated visual feature for pixel-wise alignment. Thus, the final pixel-wise alignment becomes:

$$\mathcal{L}_{pixel} = \|f_{dec}(\overline{X}_v) - X_v\|_2^2 \tag{9}$$

By introducing additional contribution of intermediate features in the pixel-wise alignment loss, we force the model to retain structural and fine-grained information in these layers and alleviate the gradual loss of visual features. This results in better-quality visual features, as evident in Fig. 1. Additionally, we note that the feature fusion and visual decoding stage only occurs during training where loss calculations are needed, and poses no additional computational overhead in inference.

# 4 EXPERIMENTS

In this section, we conduct extensive experiments from multiple perspectives to demonstrate the effectiveness of our method. First, we provide experiment settings including training data, training recipes, and evaluation benchmarks in Sec. 4.1. Next, we provide experimental results from the aforementioned benchmarks and compare with contemporary MLLMs in Sec. 4.2. EMMA surpasses other Mamba-based models on a majority of tasks and is highly competitive with transformer models of similar size. Then, we conduct a detailed inference speed comparison between EMMA and other Mamba and transformer-based models of similar size in Sec. 4.3. At last, we present ablation studies to investigate the effectiveness of our model design choices in Sec. 4.4.

## 4.1 EXPERIMENT SETTINGS

**Training Data.** Following Zhao et al. (2024a), we train EMMA on a combination of datasets consisting of LLaVA-v1.5-mixed-665k Liu et al. (2024b), LVIS-Instruct-4V Wang et al. (2023), and LRV-Instruct Liu et al. (2023b). These are conversational multi-modal and pure textual data that contain roughly 1.2 million images and dialogues.

**Backbone Models.** We select Cobra as our baseline model. Following Zhao et al. (2024a), we utilize SigLIP and DINOv2 as the visual encoder, where the final output is the concatenated feature of both models. The image decoder consists of a combination of 4 Mamba and linear layers. The MLP projector consists of stacked linear layers. We provide two versions of our model with MambaV1-2.8b Gu & Dao (2023) and MambaV2-2.7b Dao & Gu (2024) backbones.

**Training Recipes.** We directly finetune the Mamba LLM backbone, the multi-scale fusion module, the image decoder, and the MLP projector on the training data for two epochs, discarding the pretrain phase. The visual encoder is frozen at all times. We select a global batch size of 128 and a starting learning rate of 2e-5 with AdamW optimization. Our models are trained on eight 40G A100 GPUs with fully sharded data parallelism Zhao et al. (2023b).

**Evaluation Benchmarks.** We evaluate our model variants on four open-ended visual question-answer benchmarks: VQAv2 Goyal et al. (2017) and VizWiz Gurari et al. (2018) test general visual reasoning, GQA Hudson & Manning (2019) validates spatial reasoning, and TextVQA Singh et al. (2019) assesses reasoning around text. We also evaluate our models on nine comprehensive closed-set benchmarks: VSR Liu et al. (2023a) tests coarse object spatial relationships. POPE Li et al. (2023b) tests object hallucinations while HallusionBench Guan et al. (2024) assesses object illusion and visual hallucinations. MMB Liu et al. (2023c) and MME Fu et al. (2024) are both robust and holistic evaluations of MLLMs. We use reproduced results from the Cobra codebase Zhao et al. (2024a) and obtain the performance of other models in their respective papers. More details for experimental settings can be found in the Appendix.

## 4.2 EVALUATION RESULTS

We compare our model with a variety of Mamba-based and transformer-based MLLMs. These include large-scaled MLLMs: OpenFlamingo Awadalla et al. (2023), BLIP-2 Li et al. (2023a), MiniGPT-4 Zhu et al. (2023), EMU Sun et al. (2023a), EMU 2 Sun et al. (2024), InstructBLIP Wenliang et al. (2023), Shikra Chen et al. (2023a), IDEFICS Hua & Artzi (2024), Qwen-VL Bai et al. (2023), LLaVA Liu et al. (2024b), Prism Karamcheti et al. (2024), ShareGPT4V Chen et al. (2023b), MoE-LLaVA Lin et al. (2024). We also include transformers with similar scales, which encompass LLaVA-Phi Zhu et al. (2024b), MobileVLM Chu et al. (2023), MobileVLM V2 Chu et al. (2024), and TinyLLaVA Zhou et al. (2024). Lastly, we also compare with current Mamba-based MLLMs, consisting of VL Mamba Qiao et al. (2024) and Cobra Zhao et al. (2024a). We also provide the backbone LLM of each model, as well as the training data size for more fair comparison.

**Comparison with similar-scaled Mamba MLLMs.** Performance comparisons between EMMA and other Mamba-based MLLMs can be found in the bottom group of Tab. 1 as well as 6. We underline the best-performing models in this category. All models utilize a similar amount of data and Mamba backbone. We surpass the performance of Cobra, our baseline model, on every evaluated metric, demonstrating better generalizability as well as visual and spatial reasoning. The increase in performance is most apparent in MME, where there is a noticeable increase of 279 in terms of

| Model | LLM | Data Size | VQA$^{v2}$ | GQA | VizWiz | VQA$^T$ | VSR | MME | MMB |
|---|---|---|---|---|---|---|---|---|---|
| OpenFlamingo | MPT-7B | 2B | 52.7 | - | 27.5 | 33.6 | - | - | - |
| BLIP-2 | Vicuna-13B | 129M | - | 41.0 | 19.6 | 42.5 | 50.9 | 1293.8 | - |
| MiniGPT-4 | Vicuna-7B | 5M | 32.2 | - | - | - | - | 581.7 | - |
| EMU | LLaMA-13B | 3.4B | 62.0 | 46.0 | 38.3 | - | - | - | - |
| EMU 2 | LLaMA-33B | 4.1B | 84.9 | 65.1 | 54.9 | 66.6 | - | - | - |
| InstructBLIP | Vicuna-7B | 130M | - | 49.2 | 34.5 | 50.1 | 54.3 | - | 36.0 |
| InstructBLIP | Vicuna-13B | 130M | - | 49.5 | 33.4 | 50.7 | 52.1 | 1212.8 | - |
| Shikra | Vicuna-13B | 6.1M | 77.4 | - | - | - | - | - | 58.8 |
| IDEFICS | LLaMA-7B | 354M | 50.9 | - | 35.5 | 25.9 | - | - | 48.2 |
| IDEFICS | LLaMA-75B | 354M | 60.0 | - | 36.0 | 30.9 | - | - | 54.5 |
| Qwen-VL | Qwen-7B | 1.5B | 78.2 | 59.3 | 35.2 | 63.8 | - | - | 38.2 |
| LLaVA v1.5 | Vicuna-7B | 1.2M | 78.5 | 62.0 | 50.0 | 58.2 | - | 1510.7 | 64.3 |
| Prism | LLaMA-7B | 1.2M | 81.0 | 65.3 | 52.8 | 59.7 | 59.6 | - | - |
| ShareGPT4V | Vicuna-7B | 1.2M | 80.6 | 57.2 | - | - | - | 1567.4 | 68.8 |
| MoE-LLaVA | StableLM-1.6B x 4 | 2.2M | 76.7 | 60.3 | 36.2 | 50.1 | - | - | 59.4 |
| MoE-LLaVA | Phi2-2.7B x 4 | 2.2M | 77.6 | 61.4 | 43.9 | 51.4 | - | - | 65.5 |
| LLaVA-Phi | Phi2-2.7B | 1.2M | 71.4 | - | 35.9 | 48.6 | - | 1335.1 | 59.8 |
| MobileVLM-3B | MobileLLaMA-2.7B | 1.2M | - | 59.0 | - | 47.5 | - | 1288.9 | 59.6 |
| MobileVLM v2 | MobileLLaMA-2.7B | 3.6M | - | **61.1** | - | **57.5** | - | 1440.5 | 63.2 |
| TinyLLaVA | Phi2-2.7B | 1.2M | **76.6** | 60.3 | - | 51.4 | - | 1464.9 | **66.9** |
| VL-Mamba | MambaV1-2.8B | 1.2M | **76.6** | 56.2 | - | 48.9 | - | 1369.6 | 57.0 |
| Cobra | MambaV1-2.8B | 1.2M | 74.9 | 59.1 | 52.0 | 52.4 | 51.7 | 1294.3 | 51.5 |
| EMMA-V1 | MambaV1-2.8B | 1.2M | 76.3 | 60.5 | 52.1 | 57.2 | 51.5 | **1572.8** | 53.2 |
| EMMA-V2 | MambaV2-2.7B | 1.2M | 75.7 | 59.4 | **54.1** | 56.2 | **52.7** | 1454.9 | 60.8 |

Table 1: Experimental results on seven multi-modal benchmarks: VQA$^{v2}$, GQA, VizWiz, TextVQA, VSR, MME and MMB. We separate the models into three groups by horizontal lines: the first group consists of large MLLMs, the second group consists of small-scaled transformer MLLMs, and the third group consists of Mamba MLLMs. Our method surpasses other Mamba-based frameworks in all but the VQA$^{v2}$ task, where VL Mamba has a slight advantage of less than 1%. Our method also achieves competitive results with other transformer-based methods of similar scale, while attaining a much smaller computational complexity, as detailed in Sec. 4.3.

the general perception and cognition abilities of the model. There is also a 5% increase in the open-ended VQA task TextVQA, which reflects the attentiveness of fine-grained details of our model. The performance gain is least apparent in VSR due to it consisting solely of true/false answers, which are inherently coarse in nature and do not necessitate fine-grained details. Nevertheless, our model still improves upon the Cobra baseline on these benchmarks with small gains. We also outperform VL Mamba on all but the VQAv2 benchmark, where our best model underperforms by 0.3. These results suggest that Mamba MLLMs benefit from better-quality visual features through structural and hierarchical alignment, resulting in overall gains across the majority of metrics.

**Comparison with similar-scaled transformer MLLMs.** Performance comparisons between EMMA and other transformer-based MLLMs can be found in the middle group of Tab. 1. We bold-font the best-performing models for all similar-scaled MLLMs, Mamba and transformer-based. Note that although these models have similar parameter sizes to ours, their backbone LLM was trained on 1.3T (MobileLLaMA) and 1.4T (Phi-2-2.7B) textural tokens prior to the MLLM pretrain and finetune stage. On the other hand, MambaV1-2.8b was trained on the SlimPj Shen et al. (2023) and MambaV2-2.7b on the Pile Gao et al. (2020), with only 627B and 300B tokens respectively. Furthermore, MobileVLM V2 trains on nearly three times as much data as ours prior to evaluation. Other networks such as TinyLLaVA train on the more diverse ShareGPT4V dataset, which greatly improves model performance. Nevertheless, our model achieves the best VizWiz, VSR, and MME scores across all models of similar scale. Our model is also competitive on other metrics, many of which are only around 1% below the best-performing model.

**Comparison with large-scaled transformer MLLMs.** Lastly, we also compare EMMA with other large-scaled MLLMs in the upper group of Tab. 1. Although there is a significant gap in model size and training data, EMMA still achieves competitive performance in some benchmarks. We also highlight EMU and EMU2, which is a transformer MLLM that also jointly generates visual and textual tokens in an autoregressive fashion. However, both EMU and EMU2 require a sizable stable diffusion decoder to translate visual tokens into image space, along with additional training stages to optimize their latent space. These transformer-based models also demand a substantial amount of data and possess significantly larger parameter counts compared to our model. Regardless, our

model consistently outperforms EMU across all benchmarks while utilizing nearly one-fourth of its parameters. EMMA also achieves the best MME across all competing models. EMMA is second in terms of VizWiz, only 0.8 behind the best-performing model EMU2, which is nearly ten times bigger than our model. These results demonstrate the effectiveness of EMMA and show the potential of Mamba-based MLLMs as an alternative to transformer-based models.

| | EMMA-V1 | EMMA-V2 | Cobra | GPT4V | LLaVA-1.5 | Claude3 | BLIP2-T5 | Qwen-VL | MiniGPT5 |
|---|---|---|---|---|---|---|---|---|---|
| POPE | **88.0** | 87.3 | 87.2 | - | 85.9 | - | - | - | - |
| HBench | 51.0 | 47.5 | 41.4 | **65.28** | 47.1 | 56.9 | 48.1 | 39.2 | 40.3 |

Table 2: Performance comparisons for hallucination-related benchmarks on a selection of MLLMs. HBench denotes HallusionBench. Our model achieves the best performance in POPE and is closely competitive with large-sized models.

**Analysis for Hallucination.** We analyze the degree of hallucination in our models and report corresponding performances on the POPE and HallusionBench benchmark in Tab. 2. POPE concerns object hallucination which refers to generated contents inconsistent with ground-truth objects in the input, which reflect hallucination leaning towards the language modality. HallusionBench focuses on diagnosing both the visual illusion and knowledge hallucination of MLLMs. Visual illusion refers to the misinterpretation of accurate visual information, and knowledge hallucination denotes perceptions formed without relevant visual input. Thus, HallusionBench reveals hallucinations more on the effective extraction of information in the visual modality. Our model achieves the highest score on POPE, demonstrating the superior NLP capabilities of the Mamba LLM backbone. We also achieve significant gain (51.0 vs 41.4) in HallusionBench from our baseline, demonstrating the effectiveness of higher-quality visual features in reducing visual hallucinations. Our model is also competitive in HallusionBench with LLaVA-1.5 and BLIP2-T5, a 7B and 12.4B model respectively.

## 4.3 MODEL LATENCY

We evaluate model latency between EMMA and other similar-sized MLLMs, which include Cobra, TinyLLaVA, and MobileVLM V2 in Tab. 3. While VL Mamba is not open-sourced yet, we assume the latency of VL Mamba to mirror that of Cobra, as both models leverage the MambaV1-2.8b backbone architecture. All models are given the same sample image and the exact textural prompt of "Describe this image in detail". Each model is then forced to generate 256 tokens in response to this prompt for a total of 200 times, and the overall time cost is recorded as $T_{overall}$. Finally, we calculate the average time cost for each model to generate 256 tokens ($T_{avg}$) as well as the number of tokens generated per second ($N_{avg}$) as follows:

$$T_{avg} = T_{overall}/200; \quad N_{avg} = 256/T_{avg} \tag{10}$$

All evaluations are conducted on a single 40G A100 GPU. EMMA has a significant time advantage over transformer-based MLLMs and a non-trivial edge over other Mamba-based MLLMs. Mamba models, given their better theoretical guarantees, are three to five times faster than current state-of-the-art transformer-based models of similar sizes. Our model achieves even better runtime than Cobra due to more efficient processing in the MambaV2 LLM backbone.

| Model | LLM Backbone | Tokens per second | Average Time |
|---|---|---|---|
| EMMA (Ours) | MambaV2 LLM-2.7B | 149.96 | 1.71 |
| EMMA (Ours) | Mamba LLM-2.8B | 138.95 | 1.84 |
| Cobra | Mamba LLM-2.8B | 138.95 | 1.84 |
| TinyLLaVA | Phi2-2.7B | 41.46 | 6.17 |
| MobileVLM V2 | MobileLLaMA-2.7B | 39.36 | 6.50 |

Table 3: Latency table for similar-scaled MLLMs, both Mamba & transformer architectures.

## 4.4 ABLATION STUDIES

In this section, we present ablation studies for the design choices of our model in Tab. 4.

**Component-wise Ablation.** We first explore the impact of the two proposed components on the overall performance of our model. By removing the multi-scale feature fusion module, we observe

a noticeable reduction in MME, as well as slight degradation in VQAv2, GQA, POPE, and HallusionBench. This shows that the multi-scale feature fusion module plays a crucial role in enhancing performance across multiple tasks, by preventing visual information loss in the LLM, thereby enhancing the overall quality of visual features. We then remove the pixel-wise alignment loss, which is equivalent to training the plain Cobra model. We observe a substantial decline in TextVQA and HallusionBench, along with marginal decreases in GQA, VQA-V2, and MMB. Notably, TextVQA involves open-set VQA, relying heavily on intricate visual details, whereas HallusionBench addresses visual hallucinations. This outcome underscores the impact of our pixel-wise alignment loss in improving the quality of visual representations, leading to enhanced processing of fine visual details and a reduction in visual hallucinations.

| Model | VQA$^{v2}$ | GQA | VizWiz | VQA$^T$ | VSR | POPE | MME | MMB | HBench |
|---|---|---|---|---|---|---|---|---|---|
| EMMA (Ours) | **76.25** | **60.5** | 52.1 | **57.2** | 51.5 | **88.0** | **1572.8** | **53.2** | **51.0** |
| −MFF | 75.9 | 59.3 | 52.0 | 57.0 | 51.5 | 87.1 | 1294.1 | 53.1 | 50.7 |
| −PAL | 74.9 | 59.1 | 52.0 | 52.4 | **51.7** | 87.2 | 1294.3 | 51.5 | 41.4 |
| +CSM | 75.8 | 59.0 | **54.2** | 55.6 | 51.5 | 87.6 | 1420.5 | 52.7 | 50.5 |
| +AVF | 52.8 | 44.8 | 45.0 | 41.1 | 51.6 | 69.7 | 984.8 | 25.0 | 47.2 |

Table 4: Ablation table for our model. Without loss of generality, we utilize the MambaV1 version of our model for these experiments. HBench denotes HallusionBench. MFF denotes the multi-scale feature fusion module, PAL denotes the pixel-alignment loss, CSM denotes the cross-scan mechanism, and AVF denotes align visual features.

**Preserving Structural Information in MambaLLM.** We note that many works Qiao et al. (2024); Liu et al. (2024c) utilize a visual selective scan mechanism to incorporate structural information and enhance the quality of visual features. Consequently, we opt to integrate this mechanism on top of our method (before the projection MLP) to assess its impact on the quality of visual representations and model performance (+CSM). The results demonstrate a marginal change in performance, suggesting that our structural constraint framework might already suffice in generating high-quality structural visual features. The introduction of the cross-scan module may appear to be unnecessary.

**Pixel vs. Feature Alignment.** Given that the Mamba LLM receives visual features from the vision encoder, we also explore directly aligning these processed features instead of the raw image pixels (+AVF). However, the resulting performance experiences a notable decline across various metrics. We attribute this degradation to the robust structural information inherent in pixel-level images, contrasting with the extracted features from the visual encoder that may lack such details. However, despite the severe performance degradation, the model achieves a high hallusionBench score which implies the efficacy of supervising visual features in mitigating visual hallucinations.

## 5 CONCLUSION

In this study, we introduce EMMA, a novel approach designed to rectify the imbalanced quality between visual and textual latents within the MLLM framework, which adversely impacts cross-modal alignment and leads to suboptimal performance. By expanding the autoregressive generation of text tokens to operate on image-wise patches, we leverage this process as visual supervision for the visual features within the Mamba LLM. Subsequently, we introduce a pixel-wise alignment loss to align the generated visual features with the original image, thereby preserving crucial structural information. On the other hand, the absence of structural constraints in the Mamba LLM leads to a gradual loss of fine-grained details in the visual features. To address this issue and counteract the gradual information loss, we propose a multi-scale feature fusion module that hierarchically integrates multiple intermediate LLM features, effectively preserving fine-grained information across these layers. Experimental results showcase that EMMA not only significantly enhances performance across diverse benchmarks but also exhibits reduced levels of hallucination. The MambaV2 iteration of our model further reduces model latency compared to current Mamba MLLMs and achieves nearly four times the speed of similarly scaled transformer models during inference. We hope that this study opens up new ways for enhancing the visual latent quality of Mamba-based MLLMs, presenting a more efficient and performance-comparable alternative to transformer models, especially in scenarios demanding rapid responses, such as autonomous driving and robotics control.

ACKNOWLEDGMENTS

We express our gratitude to Zhiyuan Wen and Zhilin Huang for their valuable and insightful discussions during the composition of this paper. This work is partially supported by Natural Science Foundation of China under contract No. U21B2025, 62402252, and Pengcheng Laboratory Research Project No. PCL2023A08, PCL2024Y02.

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

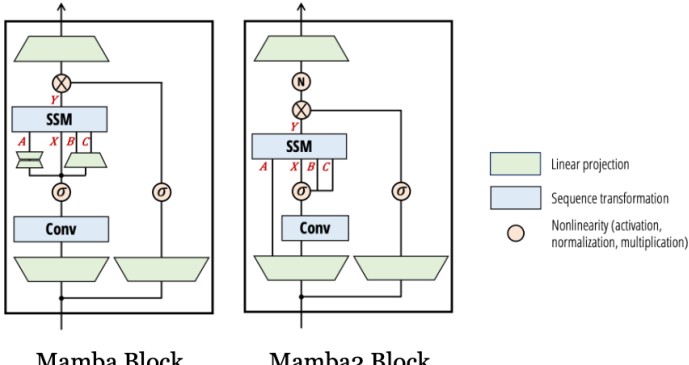

Figure 3: Comparison between Mamba and Mamba2 blocks Dao & Gu (2024).

## A   APPENDIX

### A.1   MAMBA PRELIMINARIES

Our MLLM is composed of Mamba Gu & Dao (2023) blocks, which are a form of structured state-space sequence models (S4) Gu et al. (2021a) with an input-dependent selective scanning function. We provide preliminaries regarding S4 models and the Mamba architecture in this section. Consider a sequential input $x \in \mathcal{R}^n$, the S4 model is defined by a latent state $h(t) \in \mathcal{R}^n$ with four parameters ($\Delta$, A, B, C) that maps $x$ to another sequence $y \in \mathcal{R}^n$, given by the following equation:

$$h'(t) = Ah(t) + Bx(t)$$
$$y(t) = Ch(t) \tag{11}$$

Where ($\Delta$, A, B) are continuous parameters that require discretization rules to transform into discrete parameters that enable parameter computations, given by:

$$\bar{A} = \exp(\Delta A) \quad \bar{B} = (\Delta A)^{-1}(\exp(\Delta A) - I) \cdot \Delta B$$

Then, the discretized form of S4 can be computed as follows:

$$h_t = \bar{A}h_{t-1} + \bar{B}x_t$$
$$y_t = Ch_t \tag{12}$$

This endows S4 models with beneficial properties such as resolution invariance and automatic normalization, and can be connected to gating functions in RNN. It also allows these models to be computed as equivalently a linear recurrence or a global convolution:

$$\bar{K} = \left(C\bar{B}, C\overline{AB}, \dots, C\bar{A}^k\bar{B}, \dots\right) \Leftrightarrow y = x * \overline{K} \tag{13}$$

Building upon the S4 framework, Mamba introduces a selective-scan mechanism (SSM) that is input-dependent. The main difference between S4 and Mamba is simply making the parameters ($\Delta$, B, C) as functions of the input sequence $x$, which allows the model to selectively remember or forget information depending on the current data in sequence. Due to the efficient computation for linear recurrence, Mamba is efficiently implemented and achieves significantly lower through-put time compared to transformers. Even without self-attention and MLP layers, Mamba achieves state-of-the-art performance in language, audio, and genomics tasks. Recently, there has also been ongoing research to further improve the architecture and efficiency of the Mamba layer. Dao & Gu (2024) draws the association between linear recurrence and the attention mechanism, and proposes Mamba2 with slight modifications to the Mamba architecture. Specifically, they remove the sequential linear projections and produce the SSM parameters A, B, C at the beginning of the block

instead of functions of the input $x$. They also introduce an additional normalization layer for better optimization stability. Mamba2 achieves even lower throughput than Mamba and outperforms it on language benchmarks with slightly fewer parameters. A comparison between Mamba and Mamba2 architecture is shown in Fig. 3.

## A.2  ADDITIONAL TRAINING DETAILS

In this section, we provide additional training details for our model. Given that Cobra serves as our base model, the chosen values closely mirror those employed within the Cobra framework. Model configurations and hyperparameter selection can be found in Tab. 5.

| Configuration | |
|---|---|
| Vision Encoder | DINOv2 + SigLIP ViT-SO |
| Backbone LLM | MambaV1-2.8b / MambaV2-2.7b |
| Projector | GeLU-MLP |
| Img Resolution | 384 x 384 |
| Img Tokens | 729 |
| Global Batch Size | 128 |
| Training Steps | 19K |
| Optimizer | AdamW |
| LR scheduler | Cosine Decay |
| Initial LR | 2e-5 |
| Weight Decay | 0.1 |
| Warmup Ratio | 0.03 |
| Number of Epochs | 2 |

Table 5: Detailed training configurations used for EMMA.

**Model Architecture and Hyperparameters.** We directly use off-the-shelf DINOv2 and SigLIP models as our vision encoder. The GeLU-MLP is simply a three-layer MLP with the GeLU activation function in between. For the MambaV1 model, we use the Mamba-2.8b-Zephyr version of it. The Mamba2 model is directly taken from the official github page, which is trained on the 300B Pile dataset. We select the version without attention.

**Training Data.** We provide more details regarding the training data for our model, which consists of three subsets:

1. LLaVA-Mixed-665K is a mixture of instruction-following multimodal data that contains a variety of VQA, OCR, region-level VQA, visual conversation, and language conversation data. This is the same dataset used for finetuning the LLaVA-v1.5 model Liu et al. (2024a). We provide the breakdown for this dataset as a comma-separated list with the origin and corresponding number of samples: LLaVA 158K, ShareGPT 40K, VQAv2 83K, GQA 72K, OKVQA 9K, OCRVQA 80K, A-OKVQA 66K, TextCaps 22K, RefCOCO 48K, and VG 86K, for a total of 665K data samples. According to the LLaVA-v1.5 paper, these data are preprocessed as follows: 1. For all VQA datasets, QA pairs from the same training image are merged into a single conversation. 2. For ShareGPT, they filter out invalid conversations. They also truncate long conversations that surpass 2048 tokens. 3. Each QA pair in A-OKVQA is augmented k times, where k is the number of choices per question, to counterbalance the lack of multiple-choice data. 4. They sample 80K conversations from OCRVQA. 5. For Visual Genome, they sample 10 annotations for images with additional annotations. 6. For RefCOCO, conversations are dissected into segments, each containing fewer than 10 conversations. 7. All data splits are concatenated together to form the final 665K dataset.

2. LVIS-INSTRUCT4V contains 220K pairs of image-text instructions that are generated by GPT-4v. These instructions are visually aligned and context-aware to improve cross-modal alignment. This is the dataset presented by Wang et al. (2023), in an attempt to curate a fine-grained instruction-following dataset by explicitly conditioning the generation of instruction data on both language and visual inputs. To generate this dataset, they leverage image data from LVIS Gupta et al. (2019), as well as their fine-grained annotations to prompt the GPT-4V model to undertake two key tasks: (1) generate conversational question-answer lists through self-reasoning and (2) produce high-quality image descriptions guided by precise bounding box information. To the end, they use 110K images

from LVIS and generate 220K high-quality visual instructions, which consist of 110K conversational data and 110K descriptional data. These images along with GPT-4V-generated captions form this dataset.

3. LRV-Instruction covers 16 vision-and-language tasks with open-ended instructions and answers. It includes both positive and negative instructions for more robust visual instruction tuning. This dataset Liu et al. (2023b) leverages GPT4 to cover open-ended positive and negative instructions in different linguistic styles for images in the Visual Genome dataset. They use GPT4 to create instruction-following data with the image size, bounding boxes, and dense captions as input, and generate caption instances in both declarative and interrogative formats. Afterwards, they remove instances with answers longer than 30 words and those with unneeded content. This results a total of over 400k image-visual instruction pairs after filtering.

**Data Preprocessing.** After combining these three datasets, we finetune our entire model on it. Because we utilize pre-trained vision encoders and Mamba LLM, for data pre-processing, we also adopt the same tokenizer and image transforms from the respective components. For Mamba LLM, we utilize the GPTNeoXTokenizerFast tokenizer to convert input text into textual tokens. For both the SigLIP and DinoV2 ViTs, we use their respective image transforms, as listed below: 1. SigLIP: Resize to (384, 384), center crop, normalize by mean 0.5 and std 0.5. 2. DinoV2: Resize to (384, 384), center crop, normalize by mean 0.485 and std 0.225.

## A.3 ADDITIONAL EXPERIMENTS

We present further experimental results on the following datasets to showcase the effectiveness of our method: SEED-IMG Li et al. (2024a) evaluates generative comprehension. AI2d Hiippala et al. (2021) and ScienceQA Saikh et al. (2022) both evaluate science-related topics. CCBench is a benchmark on Chinese culture Liu et al. (2023c).

| Model | SEED | ScienceQA | AI2D | CCBench |
|---|---|---|---|---|
| Cobra | 61.5 | 63.0 | 48.6 | 11.6 |
| EMMA-V1 (Ours) | 62.9 | **66.3** | **50.5** | 16.7 |
| EMMA-V2 (Ours) | **64.3** | 65.0 | 49.1 | **29.21** |

Table 6: More performance comparisons across MLLMs with similar sizes. Since none of the other 3B-scale MLLMs report performance on these benchmarks, we report the results for Cobra and ours.

As shown, both versions of EMMA achieves consistent performance gains from its baseline Cobra, demonstrating the effectiveness of pixel-wise and hierarchical alignment in mamba-based VLMs.

## A.4 ADDITIONAL EVALUATION DETAILS

In this section, we offer a concise summary of the key aspects that each benchmark emphasizes when assessing the model in order to give an overview of the strengths of our model.

**VQAv2** Goyal et al. (2017) consists of 265K image-question-answer tuples sourced from the COCO dataset Lin et al. (2014). The evaluation of models in the VQAv2 test set centers on visual recognition, visual grounding, spatial reasoning, and language comprehension.

**GQA** Hudson & Manning (2019) consists of 22M questions of daily images, where each image is associated with a scene graph of the image's objects, attributes, and relations, generated from the Visual Genome dataset Krishna et al. (2017). The GQA test set rigorously assesses the models' proficiency in visual and compositional reasoning.

**VizWiz** Gurari et al. (2018) is derived from a natural visual question-answering scenario, where visually impaired individuals captured images and verbally posed questions about them. Each image is then accompanied by 10 answers from crowd-sourcing. This benchmark tests predicting answers to visual questions and determining whether a visual question is unanswerable.

**TextVQA** Singh et al. (2019) is a multimodal dataset consisting of 45K questions on 28K images. This benchmark not only tests recognizing textual information in the given images but also reasoning over them.

**VSR** Liu et al. (2023a) comprises caption-image pairs with true/false labels. Each caption delineates the spatial relationship between two distinct objects within the image. In this task, the model is tasked with determining whether the caption accurately describes the image or not.

**MME** Fu et al. (2024) stands as a comprehensive benchmark for evaluating MLLMs, measuring both perception and cognition across a total of 14 subtasks. The perception portion includes the recognition of coarse-grained and fine-grained objects. The cognition part includes commonsense reasoning, numerical calculation, text translation, and code reasoning. When evaluating our model, we add the scores from perception and cognition tasks as our final score.

**MMB** Liu et al. (2023c) is also a comprehensive benchmark for evaluating MLLMs. It surpasses existing similar benchmarks in terms of the number and variety of evaluation questions and abilities. Answers are recorded through multiple-choice questions in both English and Chinese versions, enabling performance comparisons under a bilingual context.

**SEED** Li et al. (2024a) encompasses 19K multiple-choice questions with precise human annotations. It covers 12 evaluation dimensions, encompassing the understanding of both image and video modalities. We only select the image subset of the SEED benchmark to evaluate our models.

**ScienceQA** Saikh et al. (2022) features scientific questions and answers gleaned from academic sources. Models are evaluated on their ability to reason with scientific knowledge using questions, answer choices, and relevant contexts.

**AI2D** Hiippala et al. (2021) is a dataset comprising more than 5K grade school science diagrams with over 15K corresponding multiple-choice questions. This benchmark, like ScienceQA, also tests knowledge of MLLMs on science-related topics.

**CCBench** Liu et al. (2023c) is a part of the MMBench project that evaluates the models on the domain of Chinese Culture.

**POPE** Li et al. (2023b) targets the evaluation of object hallucination in MLLMs. Test samples include both positive and negative objects (non-existent objects), requiring models to accurately recognize positives and identify negatives. This evaluation benchmark measures hallucination more on the textual modality.

**HallusionBench** Guan et al. (2024) focuses on diagnosing both the visual illusion and knowledge hallucination of MLLMs. Visual illusion refers to the misinterpretation of accurate visual information, and knowledge hallucination denotes perceptions formed without relevant visual input. This benchmark ties more closely with testing hallucination on the visual modality.

## A.5    ADDITIONAL VISUALIZATIONS

We also present additional visualizations for intermediate features in Mamba MLLMs to demonstrate the effectiveness of our method.

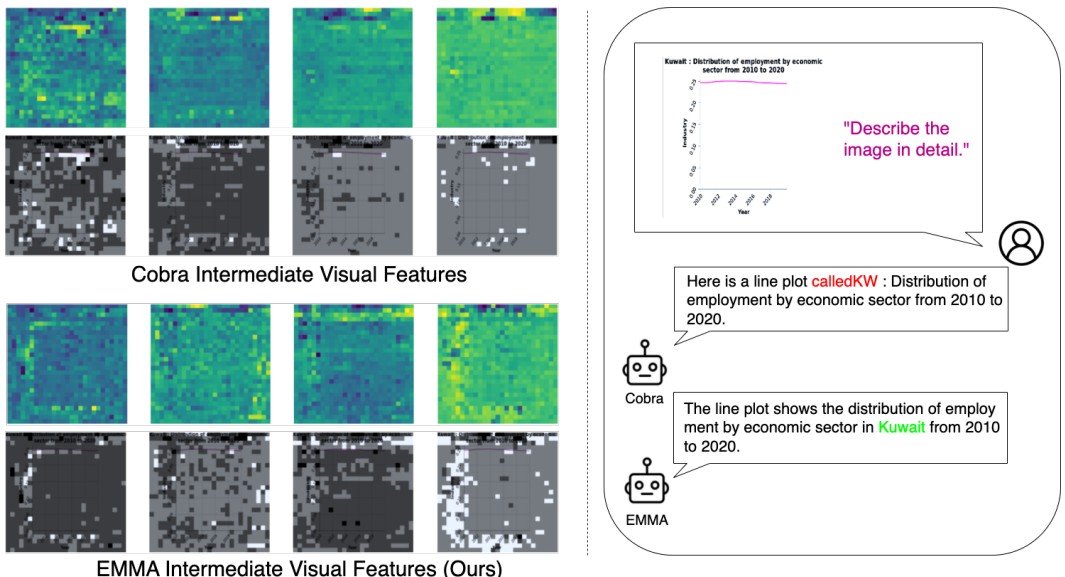

Figure 4: As shown, our method better highlights important characteristics of the image, such as the captions and horizontal and vertical axes. Cobra, on the other hand, gradually loses its visual activations on these characteristics, resulting in an incorrect reference to the plot as "calledKW".

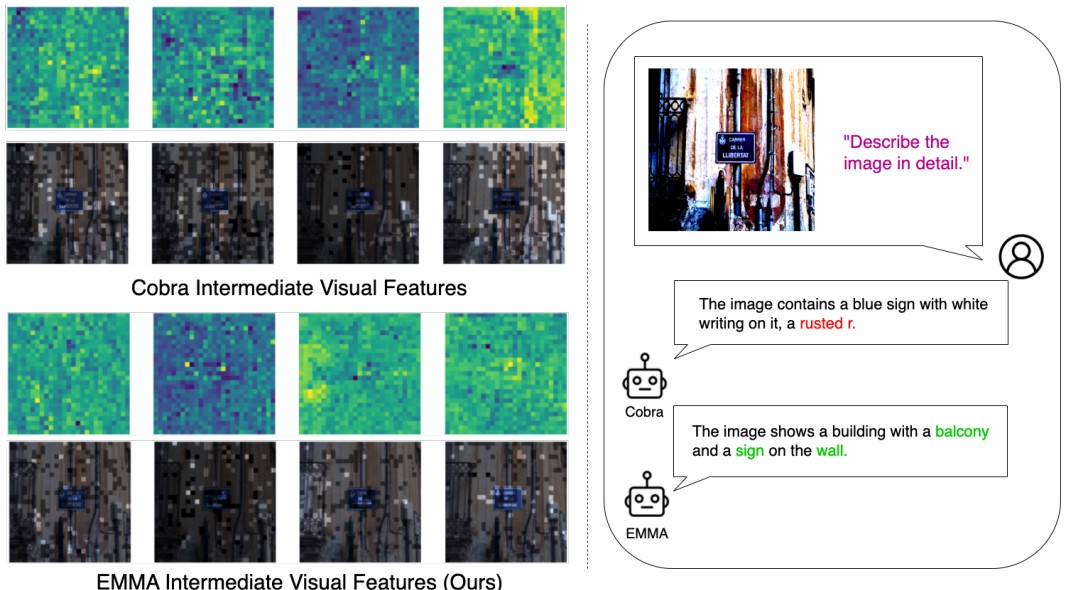

Figure 5: Similar to the previous example, EMMA is able to retain focus on important visual details even during later intermediate layers, resulting in better sensitivity to fine-grained visual details and less visual hallucination.

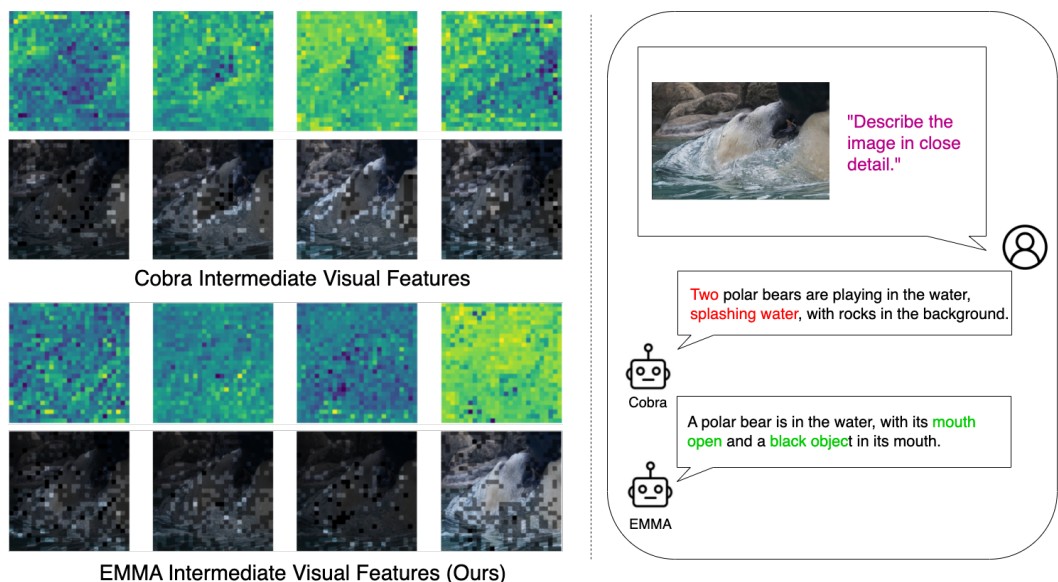

Figure 6: EMMA is also able to better grasp objects that span large areas in an image. This example showcases that EMMA is less prone to hallucinations of large objects, as opposed to Cobra.

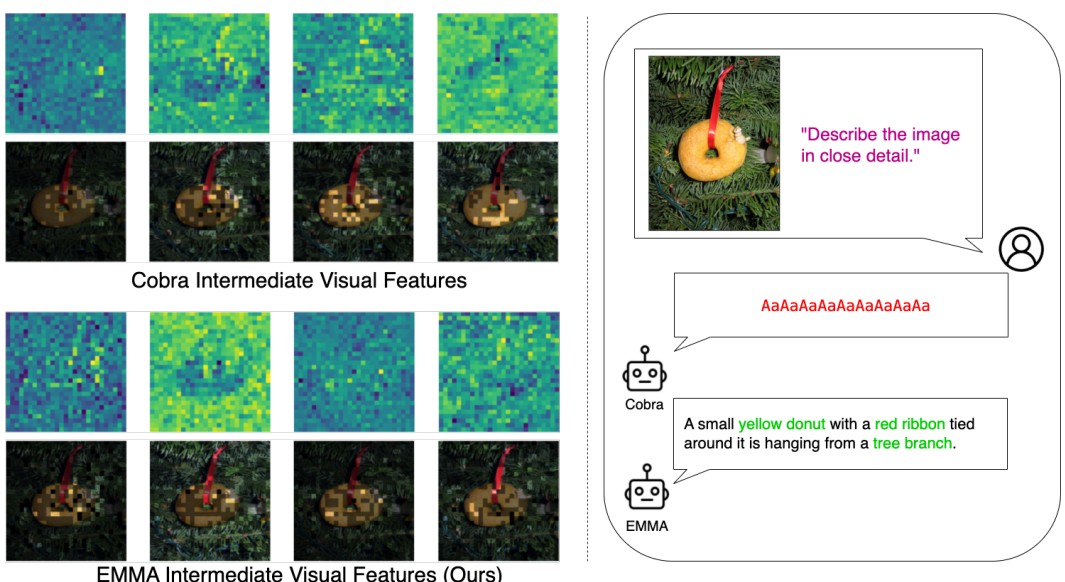

Figure 7: Another qualitative comparison between EMMA and Cobra. While Cobra fails to generate a logical answer, EMMA is able to accurately describe the image due to its ability to capture finegrained visual details.

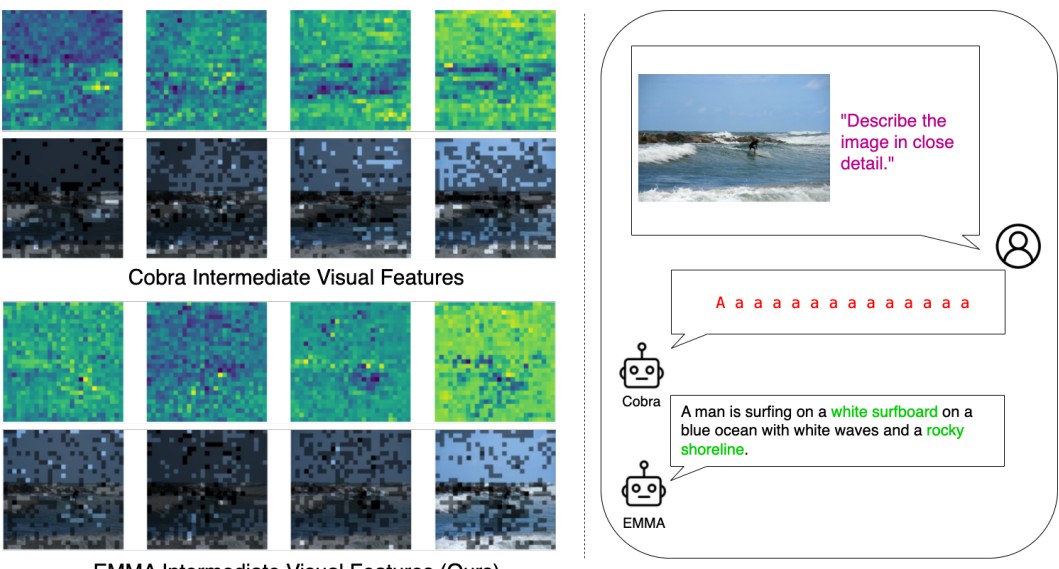

Figure 8: Another fail case for Cobra, while EMMA successfully answers the input prompt. Notice that EMMA retains the visual structure of intermediate features, where the human silhouette has been preserved.

## A.6 Additional Component Analysis

**Computation cost analysis.** We provide the following table for parameter counts of different components in our model. $Time_{Trn}$ and $Memory_{Trn}$ denote training-wise time and memory requirements, while $Time_{Eval}$ and $Memory_{Eval}$ denote evaluation-wise time and memory requirements. Decoder refers to the decoder for generating visual features for pixel-wise alignment, while MFF refers to the multi-scale feature fusion module for the hierarchical alignment. CA refers to the sum of all additional cross-attention modules introduced, to separately account for the computational complexity of cross-attention operations. Note that this row is marked gray as these parameters are also accounted for in MFF. Memory is in terms of MiB, and time is in terms of seconds. The memory of each component is calculated by

$$memory_{post-module} - memory_{pre-module}$$

The time of each component is calculated by

$$time_{post-module} - time_{pre-module}$$

The training results include additional loss calculation and optimization passes, which introduce additional time and memory usage. Note that the inference time reported here multiplied by 256 does not give the actual model inference time as presented in Table 4 (in original paper), this is due to the incorporation of cached operations during inference (which are common in all MLLM models, including transformers), which only necessitate one pass through the ViT and reduced tokens for the LLM during the text generation phase. Here we report actual processing times instead of cached times for the sake of performance measure, but note that cached times are more suitable and realistic for deployment. The following breakdown is performed uniformly with a batch size of 1, evaluated on a NVIDIA A100 GPU.

### A.6.1 Cross-Attention Analysis

**Cross-attention background and overview.** The cross attention module Lin et al. (2022) replaces key and value vectors in self-attention with inputs from different sequences, enabling the calculation of attention scores based on alternative information. Currently, cross-attention is commonly used in multimodal models for representation fusion, as evidenced by various methods from basic MLLM

| Components | $Parameters$ | $Time_{Trn}$ | $Memory_{Trn}$ | $Time_{Eval}$ | $Memory_{Eval}$ |
|---|---|---|---|---|---|
| Visual Encoder (ViT) | 731M | 2e-2 | 3,600 | 2e-2 | 3,600 |
| Projection MLP | 47M | 2e-3 | 498 | 4e-4 | 460 |
| MambaV1-2.8B LLM | 2.8B | 7e-1 | 25,266 | 7e-2 | 11,140 |
| MambaV2-2.7B LLM | 2.7B | 7e-1 | 26,210 | 6e-2 | 11,054 |
| Decoder | 174M | 5e-1 | 1,204 | N/A | N/A |
| MFF | 266M | 6e-3 | 1,346 | N/A | N/A |
| CA | 105M | 1e-3 | 892 | N/A | N/A |

Table 7: EMMA Component Analysis

approaches like Flamingo Alayrac et al. (2022) and EVLM Chen et al. (2024a) to advanced representation fusion MLLM methods such as LLAVA-UHD Guo et al. (2025) and MG-LLAVA Zhao et al. (2024b). Cross-attention has demonstrated its capabilities in fusing both same-modal and multi-modal features in these works, and is the reason why we have also opted for cross-attention.

**The use of attention-mechanisms in Mamba.** The use of the attention mechanism in Mamba-based architecture is not uncommon. The authors for the original Mamba Gu & Dao (2023) paper has provided a version of attention-augmented Mamba2 model in their follow-up work Mamba2 Dao & Gu (2024), which shows improved performances without introducing significant computational overhead. Furthermore, hybrid Mamba-transformer architectures such as Jamba Lieber et al. (2024); Team et al. (2024), Zamba Glorioso et al. (2024), and Griffin De et al. (2024) have been proposed to combine the efficiency of Mamba and state-space models with the benefits of transformer-based attentions.

**Cross-attention in EMMA.** We observe that current Mamba-based multimodal models exhibit a gradual loss of fine-grained visual cues in deeper layers of the MLLM. Thus, in an attempt to reduce this information loss, we propose to use a hierarchical alignment module to counteract this loss of information. The essence of using cross-attention in our work lies in merging information from multiple layers of representation, ensuring that information from intermediate layers is also involved in visual representation alignment to prevent loss of image information at intermediate layers. Therefore, cross-attention is not mandatory, but only serves as a means to merge layer-wise information. Thus, the replacement of it with more efficient and simpler attention mechanisms, or usage of different representation fusion modules can be explored. Nevertheless, due to the minimal number of parameters involved, there may not be a significant difference in computational efficiency, and hence not explored in this work.

As shown in Tab. 7, cross-attention modules only account for a very small proportion (roughly 3 percent) of the model's optimization parameters. Additionally, the cross-attention here is only quadratic in terms of the length of image patches and does not vary based on the length of text, hence not truly quadratic in the strictest sense. Hence the memory and time usage of the cross attention module is extremely small. Also, since the decoder and MFF modules are not utilized during inference time, they pose no additional computation cost during testing.

**Alternative Attention Mechanisms in EMMA.** As an example, we replace the cross-attention modules with lucidrains version Lucidrains (2021) of linear attention Wang et al. (2020); Katharopoulos et al. (2020a) in the EMMA-V1 model. This enables linear-wise computation of attention, and achieves the following performances:

| Model | GQA | VizWiz | $VQA^T$ | POPE | MME | MMB | HBench |
|---|---|---|---|---|---|---|---|
| EMMA-V1-CA | 60.5 | 52.1 | 57.2 | 88.0 | 1572.8 | 53.2 | 51.0 |
| EMMA-V1-LA | 60.2 | 52.1 | 57.7 | 88.1 | 1560.0 | 52.5 | 46.9 |

Table 8: Cross-attention vs. Linear Attention

EMMA-V1-CA denotes the original proposed model with cross-attention, while EMMA-V1-LA denotes EMMA but with linear attention instead. EMMA-V1-LA performs similarly to EMMA-V1-CA on the majority of benchmarks, besides HallusionBench, which experiences a 4% drop. This could be attributed to the trade-off of sub-quadratic complexity and overall performance of

linear attention. Nevertheless, EMMA-V1-LA still outperforms the baseline on HallusionBench by 5%.

### A.6.2 FEATURE-FUSION ANALYSIS

**Mamba-based feature-fusion methods.** To our knowledge, current Mamba-based feature fusion methods include MambaDFuse Li et al. (2024d), fusion-Mamba Dong et al. (2024) and Fusion-Mamba Xie et al. (2024). MambaDFuse proposes a dual-phase feature fusion module that consists of shallow fuse and deep fuse modules. The shallow fuse module achieves lightweight exchange of features from multiple modalities, while the deep fuse module, a newly designed Multi-modal Mamba (M3) block, guides the generation of modality-fused features and incorporates local detail characteristics from different modalities. Fusion-Mamba proposes a Fusion-Mamba block (FMB) to map cross-modal features into a hidden state space for interaction, thereby reducing disparities between cross-modal features and enhancing the representation consistency of fused features. FusionMamba designs a Mamba model suitable for fusion tasks by integrating the Visual State Space Model with dynamic convolution and channel-wise attention.

**Improvements from previous Mamba-based methods.** We believe the novelty of our work lies in two perspectives. First, we are the first to examine feature-fusion of Mamba models in the context of multimodal LLMs. Previous feature fusion works only focus on multi-stream multimodal models which do not utilize a Mamba-based LLM, but rather consist of separate branches with similar sizes to encode both modalities. Thus, the differences in underlying model architectures renders different design choices for feature fusion methods. Previous Mamba-based multimodal LLM methods, on the other hand, adapts the Mamba LLM to typical MLLM frameworks without considering feature fusion. Our work is the first that considers feature fusion in MLLMs to boost the quality of visual features to ensure better multimodal alignment, justified with noticeable performance gains empirically. Secondly, we are the first to examine feature-fusion of same modality features in Mamba-based models. Previous feature fusion works only consider the fusion of multi-modal features, rather than augmenting useful information of same modality features from different scales. Our work proposes a novel Mamba-based multi-scale feature fusion module that effectively combines visual features from different layers, alleviating the gradual loss of visual information in the Mamba LLM.

### A.6.3 ADDITIONAL COMPUTATIONAL COSTS ANALYSIS

**Training procedure of MLLMs.** We first present the training procedure of vision-language models (VLMs) and present the training paradigm of EMMA. Here, we are specifically referring to models inspired by the LLaVA Liu et al. (2024b) architecture, which consists of a vision encoder, a multi-modal projector and a backbone LLM, of which our work is based on. The training paradigm for LLaVA-like VLMs consists of two portions, pretraining and supervised finetuning. In the pretraining stage, all but the multimodal projector is frozen, where the projector is then trained on image-text pairs to learn the mapping from visual to token space. Then, in the supervised finetuning stage, only the vision encoder is frozen, while both the projector and backbone LLM is tuned, extending the LLM to multi-modal tasks.

**Training procedure of EMMA.** Due to the effectiveness of the Mamba LLM, Cobra discovered that Mamba-based MLLMs do not benefit much from the pretraining stage. Thus, we follow the same procedure of Cobra and discard the pretraining stage altogether, and only conduct supervised finetuning for two epochs. For the additional structures introduced, we simply add the structural and hierarchical modules to our model, and jointly optimize with the original text generation objective. This simplifies the training of our methods, where only finetuning data is used to end-to-end finetune our entire model. We also do not require a separate stage nor additional data for the training of decoder or MFF module, unlike transformer-based EMU Sun et al. (2023a) and EMU2 Sun et al. (2023b).

**Fair comparison of training costs.** We have provided the complete breakdown of training and testing costs in Tab. 7. The visual encoder has the same train-time and evaluation-time values due to it being frozen and un-optimized during either training and testing. All other components utilize less memory and time resources in evaluation, compared to training. The newly introduced structures, decoder and MFF, are discarded during testing. Note that the training time for decoder is relatively

large compared to other structures given its small size. This is due to the calculation of respective gradients that concern the update of LLM parameters due to pixel-alignment loss.

**Inference speed clarification.** Because both the structural alignment and hierarchical alignment considers visual features, which are used to boost the quality of visual latents and encourage better multi-modal alignment during the *training* stage, they are discarded during inference as they are not required for visual question answering. Thus, as shown in Tab. 7, they do not introduce additional complexity during inference. Since EMMA-V1 and Cobra uses the Mamba LLM-2.8B backbone, they possess the same inference speeds in Tab. 3.

## A.7 FURTHER CLARIFICATION OF MOTIVATION

Our research is grounded in the observation that Mamba models exhibit an imbalance in handling visual and textual information. We argue for the imbalanced quality of visual and textual latents in Mamba models through two perspectives, architecturally and empirically.

**Architectural imbalance.** The inherent linear scanning mechanism of Mamba poses challenges in effectively extracting visual features on the same level as textual features. On one hand, transformer-based Dosovitskiy (2020) approaches utilize the self-attention mechanism that extracts global relationships between different parts of the input sequence. They also utilize a position encoding for learning spatial relationships of visual tokens. On the other hand, CNN-based approaches Krizhevsky et al. (2012) utilize local receptive fields to connect each neuron to a local region of the input data, allowing interaction of neighboring pixels in all directions. The effectiveness of both methods in processing visual information lies in their ability to account for global, or to the least, local clusters of the visual input. However, Mamba utilizes a selective-scan mechanism that only processes sequences in a linear order. While Mamba models achieve excellent performances in textual data, effectively capturing long-term dependencies in multiple language benchmarks Tay et al. (2020), Mamba models possess difficulties in handling image data, where pixel-wise relationships may span multiple rows and columns. Visual Mamba models have been proposed to counteract this issue through combining additional scans from different directions Liu et al. (2024c) and utilizing a similar position encoding as ViTs Zhu et al. (2024a). However, they nevertheless still utilize the linear selective-scan mechanism. In the case where features possess large embedding dimensions, which is typical in MLLM scenarios, this becomes hard for the model to extract visual cues that are far apart spatially, as the degree of freedom between each visual token across different rows and columns become huge.

**Empirical imbalance.** Mamba models experiences an imbalance in the empirical evaluation of vision tasks and textual tasks, when scaled up in parameters. Ren et al. (2024) observes the performance degradation of visual Mamba models Xu et al. (2024b) when scaled up. Below is a table taken directly from their paper.

| Model | VIM-B | VIM-L | VIM-H |
|---|---|---|---|
| Size (M) | 98 | 340 | 755 |
| ImageNet Acc. | 81.2 | 81.0 | Collapsed |

Table 9: Degradation of visual Mamba models when scaled up, sourced from Ren et al. (2024).

As shown, plain Mamba models experience a severe degradation in processing visual features when scaled up. In the 700M parameter-scale, the visual Mamba collapses entirely, rendering the need for effective approaches to alleviate this performance degradation, specifically in Mamba-based models. On the other hand, plain Mamba models do not experience performance degradation in processing textual data, as shown from the experiment results in the original Mamba papers Gu & Dao (2023); Dao & Gu (2024).

| Model | Mamba-790M | Mamba-1.4B | Mamba-2.8B | Mamba2-780M | Mamba2-1.3B | Mamba2-2.7B |
|---|---|---|---|---|---|---|
| Size (M) | 790 | 1,400 | 2,800 | 780 | 1,300 | 2,700 |
| HellaSwag | 55.1 | 59.1 | 66.1 | 54.9 | 59.9 | 66.6 |
| LAMBADA | 62.7 | 64.9 | 69.2 | 61.7 | 65.7 | 69.7 |

Table 10: Performance of textual Mamba models when scaled up, taken from Gu & Dao (2023); Dao & Gu (2024).

Thus, we observe an imbalance of the processing capabilities of Mamba on visual and textual inputs, when scaled up in billion-parameter models. This phenomenon renders the need for more effective visual processing in large-scaled Mamba models to ensure that visual and textual features are adequately aligned.

**Motivation behind loss of fine-grained visual cues.** We discovered the problem of the loss of fine-grained visual cues when directly extracting and comparing intermediate features of the Cobra model Zhao et al. (2024a). Specifically, we find that visual features in deeper layers of the Mamba LLM exhibit increased portions of noise (manifested in unnatural specs in the image) and loss of structural integrity (manifested in the loss of topological structures) in the Cobra baseline model, which can be found in our visualizations in the main paper. Subsequently, we designed the hierarchical alignment loss to enhance the participation of intermediate visual features to alleviate the gradual loss of visual information. Qualitative examples are shown in Appx. A.5.

### A.7.1 SIGNIFICANCE OF EMMA

**Application of Mamba in MLLMs.** To deploy Mamba models in MLLM settings, where large-scaled Mamba models are necessary in processing both textual and visual information, it is crucial to preserve a balance in the quality of visual and textual features to achieve multi-modal alignment. Previous Mamba MLLM works such as VL Mamba and Cobra directly replaces the transformer LLM with the Mamba LLM model without addressing this imbalance of feature quality. Still, the performance of these models show the potential of extending mambas to the realm of MLLMs, especially with their fast inference speeds.

**Ineffectiveness of transformer-based techniques.** Furthermore, as a preliminary study for this work, we have also tried transformer-based multimodal alignment approaches that enhance visual features or encourage multimodal alignment, such as visual-feature compression Li et al. (2024c), masked-alignment Mizrahi et al. (2024); Bitton et al. (2021), and contrastive losses Shen et al. (2021a); Radford et al. (2021), all of which fail in the Mamba LLM setting. Here, we show the respective model performances on the TextVQA dataset, which we find is a good indicator for overall model performance.

| Model | EMMA-V1 | EMMA-V2 | Cobra | Compression | Masked | Contrastive |
|---|---|---|---|---|---|---|
| TextVQA | 57.2 | 56.2 | 52.4 | 45.8 | 46.1 | 50.2 |

Table 11: Ablation with current transformer-based approaches on TextVQA

As shown, transformer-based approaches tailored for multimodal alignment in MLLM settings may in fact negatively impact model performance, which renders the necessity to devise Mamba-specific multi-modal alignment approaches.

**Novelty of our approach.** Autoregressive pretraining on the visual features are shown to improve visual Mamba models Ren et al. (2024), and for the first time allows visual Mamba models to scale up to large-sized models (600M). Given the effectiveness of autoregressive pretraining in promoting the single-modal visual Mamba model, we extend this approach to the multimodal LLM setting. Subsequently, we propose the pixel-wise alignment loss to encourage the generated autoregressive features to structurally align with the input image. Furthermore, when visualizing the LLM features, we notice a gradual decline of the intermediate visual features, causing the final generated visual feature to degrade. In response, we propose the multi-scale feature fusion module to tackle this issue, leading to better preservation of intermediate visual features, inducing better multimodal alignment of visual and textual features. In essence, our method specifically tackles the issue of multi-modal alignment in Mamba models, while the direct application of transformer-based multi-modal methods have failed. As a result, EMMA demonstrates improved performances on a variety of metrics, and most importantly, significantly reduces the degree of visual hallucinations in the HallusionBench benchmark, surpassing transformer models with twice the size. We believe this study provides groundwork for further research of Mamba models in the MLLM framework as a potential replacement for transformer-based approaches, with their comparable performance and superior inference latency.

**Potential impact and application of our observations and techniques.** On one hand, we believe the insights introduced in this paper will help inspire future research of mamba models in the realm

of MLLM and other downstream tasks. Specifically, our approach to strengthen the quality of visual features in large-scaled mamba models may provide future researchers with a framework for multimodal alignment in Mamba models. Our light-weight Mamba-based decoder also serves as an efficient means to conduct feature reconstruction in the visual modality. Furthermore, our multiscale feature fusion module can be an inspiration for combining same-modality features in future Mamba-based frameworks. On the other hand, we aim for our research to pitch Mamba as a potential contender among transformer-based architectures. Given Mamba's swift inference capabilities, we show in this work that Mamba-based MLLMs can achieve competitive performance with transformer models, and underscores its potential for widespread adoption and integration into diverse AI applications.

