# OpenReview forum: "EMMA: Empowering Multi-modal Mamba with Structural and Hierarchical Alignment"
_ICLR.cc/2025/Conference — ICLR 2025 Poster_

### Official Review · Reviewer_yy1F · 2024-10-28

**Soundness:** 3
**Presentation:** 3
**Contribution:** 3
**Rating:** 6
**Confidence:** 4

**Summary:**

Mamba-based architectures are promising for deep learning models but current Mamba multi-modal large language models (MLLM) are insufficient in extracting visual features. This paper proposes Empowering Multi-modal Mamba with Structural and Hierarchical Alignment (EMMA). It includes a pixel-wise alignment module for structural alignment at the image level to extract fine-grained visual information autoregressively. Additionally, a multi-scale feature fusion (MFF) module is proposed for hierarchical alignment at the feature level to prevent the degradation of visual information. Extensive experiments on various multi-modal benchmarks show that the model has lower latency than other Mamba-based MLLMs and is nearly four times faster than transformer-based MLLMs of similar scale during inference.

**Strengths:**

1. The imbalance in the quality of visual and texture latent in the Mamba-based VLM does make sense to investigate and this paper proposes an effective method, named Empowering Multi-modal Mamba with Structural and Hierarchical Alignment (EMMA) to solve this problem.

2. The experiments are sufficient to verify the effectiveness of the proposed method.

3. The paper is well-written and easy to follow.

**Weaknesses:**

1. EMMA essentially enhances the visual representation ability of the model. Therefore, the comparison with peers should be conducted, e.g. the contrastive learning in CLIP and SigLIP. Besides, does the masked image construction loss achieve a function similar to EMMA?

2. EMMA uses the visual features from the vision encoder as the target of the Pixel-wise Alignment Loss. However, visual features from the vision encoder may lose the fine-grained information. Does this problem affect the performance of EMMA?

3. High-resolution image is an important direction for MLLM. Structural constraints on the high-resolution visual features are an interesting point to discuss. Authors are suggested to conduct the experiment under a high-resolution setting.

4. Authors are suggested to display more qualitative results.

**Questions:**

Please see the weaknesses.

---

> ### Author Response · Authors · 2024-11-21
> **Q1. EMMA essentially enhances the visual representation ability of the model. Therefore, the comparison with peers should be conducted, e.g. the contrastive learning in CLIP and SigLIP.**
>
> **Q1. EMMA essentially enhances the visual representation ability of the model. Therefore, the comparison with peers should be conducted, e.g. the contrastive learning in CLIP and SigLIP.**
>
> We believe that EMMA and CLIP-related models cannot be adequately compared due to their structural and architectural differences. We explain our reasoning in the following paragraphs.
>
> **Model architecture for EMMA.** EMMA is an instruction-tuned large multimodal model for general-purpose visual and language understanding. It is architecturally similar to its MLLM transformer counterparts, such as LLaVA [1], which mostly consists of a ViT, MLP projector, and an LLM backbone. Given an image and a corresponding textual prompt, EMMA is trained to generate a corresponding textual response (and an image response).
>
> **Difference between CLIP models and MLLM models.** CLIP [2] and SigLIP [3] models, on the other hand, belong to the class of contrastive language-image pretraining models, which consist of an image encoder and a text encoder. Unlike instruction-tuned large multimodal models, which utilize the large language model to process features from both modalities and output textual responses, CLIP and related models encode both modalities separately and only output probability scores that reflect the alignment between the input image and text. Subsequently, CLIP and related models are mostly used in retrieval tasks, while EMMA and LLaVA are used in visual question-answering tasks. Because they possess different underlying structures and produce different forms of output, they cannot be equally compared in this manuscript and can be investigated in future work.

---

> > ### Comment · Reviewer_yy1F · 2024-11-24
> > **Further discussion on Q1**
> >
> > Thanks for your reply! Maybe I don't express my concerns for Q1 clearly. My apologizes. Here I further express my concerns.
> >
> > Q1: Pixel-wise alignment loss in your is essentially proposed to enhance the visual representation ability of the model. The contrastive loss in CLIP and SigLIP is also proposed to enhance the visual representation. I am curious about the comparison experiments between them, i.e., replacing the pixel-wise alignment loss with that in CLIP or SigLIP directly, and other components in your method are not required to be changed.
> >
> > Hope further clarification can help you understand my concerns.

---

> ### Author Response · Authors · 2024-11-21
> **Q2. Besides, does the masked image construction loss achieve a function similar to EMMA?**
>
> **Q2. Besides, does the masked image construction loss achieve a function similar to EMMA?**
>
> **Difference between pixel-wise alignment loss and masked image construction loss.** We believe that the masked image reconstruction loss, which is presented in masked autoencoders [4] does not achieve similar results to EMMA. We analyze this in two aspects: form-wise, and training-wise.
>
> Form-wise, masked reconstruction loss $\mathcal{L}_{mae}$ takes the form
>
> $$\mathcal{L}_{mae}(Image\_{masked}, \mathrm{MAE}(Image\_{masked}))$$
>
> whereas our pixel-alignment loss $\mathcal{L}_{pix}$ takes the form
>
> $$\mathcal{L}_{pix}(Image\_{full}, \mathrm{EMMA}(Text\_{full}, Image\_{full}))$$
>
> $\text{MAE}$ consists of an MAE encoder and an MAE decoder and can be expanded as
>
> $$\mathrm{MAE}(Image\_{masked}) = MAE_{decoder} (MAE_{encoder}(Image\_{masked}))$$
>
> On the other hand, $EMMA$ consists of a ViT vision encoder, an MLP projector, a Mamba LLM backbone, and a Mamba-based decoder, and can be expanded as
>
> $$\mathrm{EMMA}(Text\_{full}, Image\_{full}) = EMMA_{decoder} (LLM_{Mamba}(MLP_{Proj}( ViT(Text\_{full})), Image\_{full}))$$
>
> We make two observations: Firstly, the inputs differ, where the pixel-alignment loss aligns full images with reconstructed full images, while MAE loss aligns masked images with reconstructed full images. Secondly, the model architectures differ, where the pixel-alignment loss operates on a MLLM structure that also incorporates textual information for the reconstruction of images, while the MAE loss operates on a simple visual encoder-decoder structure.
>
> Training-wise, masked reconstruction loss is first pre-trained on masked-out images and then finetuned on specific downstream datasets (with unmasked full image patches) for final evaluation. This introduces additional training steps and requires more training data. On the other hand, EMMA completely skips the pretraining phase and directly adds in the pixel-wise alignment objective during the finetuning process, requiring no additional training stages and data for the joint optimization of pixel-wise alignment loss and text autoregressive loss.

---

> > ### Comment · Reviewer_yy1F · 2024-11-24
> > **Discussion on Q2**
> >
> > Thanks for your clarification. Is it possible to conduct the comparison experiments between them?

---

> ### Author Response · Authors · 2024-11-21
> **Q3. EMMA uses the visual features from the vision encoder as the target of the Pixel-wise Alignment Loss. However, visual features from the vision encoder may lose the fine-grained information. Does this problem affect the performance of EMMA?**
>
> **Q3. EMMA uses the visual features from the vision encoder as the target of the Pixel-wise Alignment Loss. However, visual features from the vision encoder may lose the fine-grained information. Does this problem affect the performance of EMMA?**
>
> EMMA uses the original image and the visual portion of the LLM output for pixel-wise alignment. Your insight is completely correct – if we replace the original image with visual features from the vision encoder to conduct pixel-wise alignment, where the LLM visual features are aligned with the visual features from the vision encoder, this results in severe performance degradation and negatively affects the performance of EMMA, presumably due to the loss of fine-grained information, as you pointed out. The ablation study in section 4.4 Pixel vs. Feature Alignment showcases this problem. Thus, to mitigate the loss of fine-grained visual information, we align the original image with the LLM visual feature in a pixel-wise fashion, thus the name pixel-wise alignment loss.

---

> > ### Comment · Reviewer_yy1F · 2024-11-24
> > **Discussion on Q3**
> >
> > Thank you for your clarification.

---

> ### Author Response · Authors · 2024-11-21
> **Q4. High-resolution image is an important direction for MLLM. Structural constraints on the high-resolution visual features are an interesting point to discuss. Authors are suggested to conduct the experiment under a high-resolution setting.**
>
> **Q4. High-resolution image is an important direction for MLLM. Structural constraints on the high-resolution visual features are an interesting point to discuss. Authors are suggested to conduct the experiment under a high-resolution setting.**
>
>
> **Current high-resolution benchmarks.** Thank you for bringing this issue to our attention. For our benchmark performance report, we only conducted experiments on commonly-used VQA benchmarks from previous MLLM works. Subsequently, we have researched the current literature for high-resolution benchmarks. [5] proposes a high-resolution benchmark for multimodal models named MagnifierBench, that tests the ability of the model to discern the details of small objects in high-resolution input images. However, this dataset remains unavailable to public, where their huggingface repo link returns a 404 error. (Please see https://github.com/Luodian/Otter/issues/343). HRVQA [6] is another benchmark proposed to evaluate the understanding capability of VQA models for aerial images in 1024 x 1024 resolution. However, their benchmark requires an evaluation server to evaluate the model predictions, which remain unavailable on their official project page (https://hrvqa.nl/). Consequently, we are unable to find any high-resolution VQA benchmarks that suit the evaluation of MLLMs. If you have any high-resolution VQA benchmarks for multimodal LLMs in mind, please let us know and we will test our model at once.

---

> > ### Comment · Reviewer_yy1F · 2024-11-24
> > **Discussion on Q4 and Q5**
> >
> > Q4: Thank you for expressing your concerns. The high-resolution setting can be referred to Mini-Gemini [1] and TokenPacker [2].
> >
> > Q5: Thanks for providing additional results. Authors are suggested to highlight the modification in different color, e.g., blue or red, in the revision.
> >
> > [1] Li Y, Zhang Y, Wang C, et al. Mini-gemini: Mining the potential of multi-modality vision language models[J]. arXiv preprint arXiv:2403.18814, 2024.
> >
> > [2] Li W, Yuan Y, Liu J, et al. Tokenpacker: Efficient visual projector for multimodal llm[J]. arXiv preprint arXiv:2407.02392, 2024.
> >
> > **Justification for my rating**
> >
> > There are still some concerns remain. I hope to see the authors' responses to my questions. I will further consider raising my score if all my concerns are resolved. In the current stage, I am unable to give a higher score for this paper.

---

> > > ### Author Response · Authors · 2024-11-25
> > > **Further Discussion on Q4 and Q5**
> > >
> > > Thank you again for your reply to our comments.
> > >
> > > Q4. Thank you for your further clarifications.
> > > **Difficulties in making fair comparisons.** We believe that with the current architecture of our model, fair comparisons with these methods in high resolution are hard to achieve. Mini-Gemini proposes a dual vision encoder that contains an adaptive CNN-based high-resolution (HR) encoder and a transformer-based low-resolution (LR) encoder. The high-resolution features are obtained by upsampling and concatenating features from different convolution stages, which are unconstrained by input shape. On the other hand, Token Packer proposes a dynamic image slicing scheme to flexibly handle higher resolution images to convert into their ViT-friendly 336x336 dimensions. A commonality of their approaches is that they allow **adaptable image sizes** either through more flexible convolution layers or through dynamic slicing. On the other hand, EMMA utilizes a fixed DinoV2 and SigLIP visual encoder (both take 336x336 pixels). We also do not include an adaptive slicing technique as designed by TokenPacker. Thus, it becomes extremely difficult to evaluate high-resolution images in our current setting, which requires a complete re-training of our model through a different adaptive visual pipeline.
> > >
> > > **Sensitivity to fine-grained detail vs. high resolution.** The inspiration behind proposing the multi-scale fusion module for preserving fine-grained detail comes from the observation that Mamba models tend to gradually lose fine-grained visual information. Thus, our model is mostly concerned with reducing this information loss in the general image domain and not about image resolution. In fact, the degradation of visual features is more significant in lower resolutions and better reflects the capability of our model to retain fine-grained visual features without losing them.
> > >
> > > **Issue with current high-resolution benchmarks.** Current methods utilize the same low-resolution VQA tasks and simply upsample the images as high-resolution. These images are still low-resolution to begin with, and may still fail to contain fine-grained visual information even when upsampled to high resolutions. We sample a range of images from the GQA, VQA-v2, TextVQA, VSR, and POPE benchmarks and show that the images are (333, 500), (640, 480), (943, 1024), (480, 640), (426, 640) in dimension. These resolutions, besides the images from the TextVQA dataset, are far from the reported upsampled 1088, 1344, and 1536 dimensions. Thus, we believe that naturally high-resolution images are needed to fully validate the performance of high-resolution images.
> > >
> > > **Attempt to show results in high-resolution.** Given the limited time and computation budget, we will try our best to show our model performances on high-resolution images. Because the resolution for our model is fixed, due to the nature of frozen ViT, a quick solution could entail slicing a high-resolution image into a fixed number of slices and taking the average performance on all such slices as the final result. We will use the remaining time to perform this experiment and update on OpenReview as soon as possible. In the case where we do not finish in time, we will continue running these experiments and include them in the final accepted version of the manuscript.

---

> ### Author Response · Authors · 2024-11-21
> **Q5. Authors are suggested to display more qualitative results.**
>
> **Q5. Authors are suggested to display more qualitative results.**
>
> We have included 3 more qualitative results showcasing the visualization of intermediate features and model responses, for a total of 5 visualizations in the appendix of the paper (we have resubmitted the new revised paper to openreview). These visualizations demonstrate that EMMA is able to retain focus on important visual details even during later intermediate layers, resulting in better sensitivity to fine-grained visual details and less visual hallucination.

---

> ### Author Response · Authors · 2024-11-21
> **References**
>
> 1. Haotian Liu, Chunyuan Li, Qingyang Wu, and Yong Jae Lee. Visual instruction tuning. Advances
> in neural information processing systems, 36, 2024
> 2. Alec Radford, Jong Wook Kim, Chris Hallacy, Aditya Ramesh, Gabriel Goh, Sandhini Agarwal, Girish Sastry, Amanda Askell, Pamela Mishkin, Jack Clark, et al. Learning transferable visual models from natural language supervision. In International conference on machine learning, pp. 8748–8763. PMLR, 2021.
> 3. Xiaohua Zhai, Basil Mustafa, Alexander Kolesnikov, and Lucas Beyer. Sigmoid loss for language image pre-training. In Proceedings of the IEEE/CVF International Conference on Computer Vision, pp. 11975–11986, 2023.
> 4. Kaiming He, Xinlei Chen, Saining Xie, Yanghao Li, Piotr Doll´ar, and Ross Girshick. Masked autoencoders are scalable vision learners. In Proceedings of the IEEE/CVF conference on computer vision and pattern recognition, pp. 16000–16009, 2022.
> 5. Bo Li, Peiyuan Zhang, Jingkang Yang, Yuanhan Zhang, Fanyi Pu, and Ziwei Liu. Otterhd: A high-resolution multi-modality model. arXiv preprint arXiv:2311.04219, 2023.
> 6. Kun Li, George Vosselman, and Michael Ying Yang. Hrvqa: A visual question answering benchmark for high-resolution aerial images. ISPRS Journal of Photogrammetry and Remote Sensing, 214:65–81, 2024.

---

> ### Author Response · Authors · 2024-11-24
> **Looking forward to further discussions!**
>
> Dear Reviewer yy1F,
>
> Thank you for your insightful comments. We were wondering if our response and revision have resolved your concerns. We have attempted to address your initial questions through our replies and are eager to clarify any further points you might raise. Please feel free to provide additional feedback. We greatly appreciate your continued engagement.
>
> Best regards, Authors

---

> ### Author Response · Authors · 2024-11-24
> **Reply to Further discussion on Q1**
>
> Thank you for the clarification! Due to limited time and computational resources, we might not be able to conduct a full range of experiments on all metrics. However, as a preliminary study for this paper, we have actually tried directly applying the CLIP loss to our baseline model, where the CLIP loss aligns the textual and visual embeddings in the mamba LLM. However, we observe that the contrastive loss, as well as other transformer-based techniques for multimodal alignment, fails to offer improvement in the scenario of Mamba-based LLMs, as evidenced in their performance in TextVQA (See *Contrastive* in the below table):
>
> | Model        | EMMA-V1 | EMMA-V2 | Cobra | Compression | Masked | Contrastive |
> |--------------|---------|---------|-------|-------------|--------|-------------|
> | TextVQA      | 57.2    | 56.2    | 52.4  | 45.8        | 46.1   | 50.2        |
>
> Here Compression denotes visual feature compression techniques (TokenPacker), Masked denotes Masked Language Modeling, and Contrastive denotes contrastive vision-language loss as in CLIP. As shown, using a CLIP loss reduces the baseline Cobra model from 52.4 to 50.2.
>
> **Hypothesis on CLIP loss performance in Mamba MLLMs.** We hypothesize that the reason behind this phenomenon is the fact that CLIP only strengthens the alignment between textual and visual features rather than boosting the quality of visual features, such as EMMA. Due to the inherent imbalance of visual and textual latents of Mamba models, this results in further reduced performance, where high-quality textual features are further aligned with low-quality image features.
>
> In the final accepted version of the paper, we will include more experiments detailing this issue, as a new section in the appendix.

---

> ### Author Response · Authors · 2024-11-25
> **Further discussion for Q2.**
>
> As a preliminary study for this paper, we have experimented with masked language modeling, where certain text tokens have been masked and predicted by the model. It can be seen as a text-based version of masked reconstruction loss. However, we observe that this approach also reduces performance (as shown in the Table in **Further discussion on Q1**). Thus, we have resorted to alternative design choices when designing EMMA. However, we would assume that using MAE on Mamba-based MLLMs would further reduce performance. Since Mamba MLLMs already experience degradation on text-wise masked reconstruction, their image counterparts may experience further degradation, as Mamba models struggle even further with visual latents.
>
> **Difference between MAE and pixel-alignment loss in terms of effectiveness.** We argue that our pixel-alignment loss is a more general version of masked image construction loss. Due to the linear scanning mechanism of Mamba, which processes inputs in sequential order, masking the input at certain tokens is equivalent to utilizing the previous token to predict the masked token autoregressively. Thus, pixel-alignment loss can be seen as a more general form of masked image construction loss where all pixels have been masked and require prediction. This results in requiring less ground-truth knowledge of pixel values compared to MAE loss, which in turn translates to better generalizability and adaptability. On the other hand, by masking out random pixels, MAE may accidentally damage the overall structure of the image when key pixels in the image are lost due to random masking. This results in worsened multimodal alignment as the overall structure is not preserved for visual supervision in Mamba LLMs.
>
> We will try our best to conduct further experimentation regarding this issue, however, due to limited time and computational resources, we may not finish in time. However, we will continue running them and include them in the appendix of the final version of this paper if accepted.

---

> ### Comment · Reviewer_yy1F · 2024-11-25
>
> Thank you for your responses. I have no further concerns and will maintain the initial score (6: borderline accept) for your paper.

---

> > ### Author Response · Authors · 2024-11-25
> > **Thank you for your comments!**
> >
> > Thank you again for your comments and suggestions for our paper. We greatly appreciate your input to our paper with respect to additional analysis of losses and incorporation of high-resolution dataset results. We will make sure to further polish our paper based on these aspects in the final version, if accepted to the conference.

---

### Official Review · Reviewer_T8Kc · 2024-11-02

**Soundness:** 3
**Presentation:** 3
**Contribution:** 3
**Rating:** 6
**Confidence:** 4

**Summary:**

The paper introduces Empowering Multi-modal Mamba with Structural and Hierarchical Alignment (EMMA), enhancing Mamba multi-modal large language models (MLLM) by improving their ability to extract fine-grained visual information. EMMA uses a pixel-wise alignment module for better structural alignment and a multi-scale feature fusion (MFF) module to maintain visual information during cross-modal alignment. Extensive experiments demonstrate that EMMA achieves lower latency and faster performance compared to other Mamba-based MLLMs, with superior cross-modal alignment and reduced hallucination. The code for this model will be made available.

**Strengths:**

1. Well Written: The paper is clearly articulated, making the complex concepts accessible and easy to understand for the reader.

2. Motivation Well Clarified: The authors firstly identify the imbalance in the quality of visual and textual latent features in MLLM Mamba models. They propose a pixel-wise alignment approach to autoregressively enhance the learning and processing of structured visual features, leading to improved cross-modal alignment between visual and textual latents.

3. Experiments Solid: The paper presents comprehensive experiments conducted on a variety of multi-modal benchmarks. These experiments include thorough comparisons with current state-of-the-art models based on both Mamba and transformer architectures, demonstrating the robustness of the proposed approach.

**Weaknesses:**

1. Why is there an imbalance in the quality of visual and textual latents in the Mamba LLM, where coarse visual features are ineffectively aligned with higher-quality textual features? How does this differ from transformer-based models? Do transformer-based models face the same issue?

2. Could you provide a more objective, quantitative analysis of the loss of fine-grained visual cues in the LLM, as the current visualizations with just a few images seem insufficient? Is this phenomenon common across MLLMs, or is it specific to the use of the Mamba LLM?

3. Why do the current results show that the Mamba LLM still lags behind the transformer architecture? In this case, does the speed advantage gained by using the Mamba architecture sufficiently compensate for the performance loss?

**Questions:**

1. Currently, models use ViT as the image encoder. Is it possible to build an MLLM entirely based on Mamba? What potential advantages and challenges might this approach entail?

2. In transformer-based LLMs, is there a similar issue with pixel-level alignment? Would the method proposed in the paper also be applicable to transformer architectures?

---

> ### Author Response · Authors · 2024-11-21
> **Q1. Why is there an imbalance in the quality of visual and textual latents in the Mamba LLM, where coarse visual features are ineffectively aligned with higher-quality textual features? How does this differ from transformer-based models? Do transformer-based models face the same issue?**
>
> **Q1. Why is there an imbalance in the quality of visual and textual latents in the Mamba LLM, where coarse visual features are ineffectively aligned with higher-quality textual features? How does this differ from transformer-based models? Do transformer-based models face the same issue?**
>
> A short answer to your question is: transformer models also experience this issue, however, visual Mamba models generally experience significantly worse performance degradations when scaling-up, rendering the need for effective visual processing in large-scaled Mamba models.
>
> **Imbalance in the quality of visual and textual latents in Mamba models.** We argue for the imbalanced quality of visual and textual latents in Mamba models through two perspectives, architecturally and empirically.
>
> **Architectural imbalance.** The inherent linear scanning mechanism of Mamba poses challenges in effectively extracting visual features on the same level as textual features. On one hand, transformer-based [1] approaches utilize the self-attention mechanism that extracts global relationships between different parts of the input sequence. They also utilize a position encoding for learning spatial relationships of visual tokens. On the other hand, CNN-based approaches [2] utilize local receptive fields to connect each neuron to a local region of the input data, allowing interaction of neighboring pixels in all directions. The effectiveness of both methods in processing visual information lies in their ability to account for global, or to the least, local clusters of the visual input. However, Mamba utilizes a selective-scan mechanism that only processes sequences in a linear order. While Mamba models achieve excellent performances in textual data, effectively capturing long-term dependencies in multiple language benchmarks [3], Mamba models possess difficulties in handling image data, where pixel-wise relationships may span multiple rows and columns. Visual Mamba models have been proposed to counteract this issue through combining additional scans from different directions [4] and utilizing a similar position encoding as ViTs [5]. However, they nevertheless still utilize the linear selective-scan mechanism. In the case where features possess large embedding dimensions, which is typical in MLLM scenarios, this becomes hard for the model to extract visual cues that are far apart spatially, as the degree of freedom between each visual token across different rows and columns become huge.
>
> **Empirical Imbalance.** Our work is inspired by [6] that observes the performance degradation of visual Mamba models when scaled up. Below is a table taken directly from their paper [7, 8].
>
> | Model        | VIM-B | VIM-L | VIM-H |
> |--------------|-------|-------|-------|
> | Size (M)     | 98    | 340   | 755   |
> | ImageNet Acc.| 81.2  | 81.0  | Collapsed |
>
>
> As shown, plain Mamba models experience a severe degradation in processing visual features when scaled up. In the 700M parameter-scale, the visual Mamba collapses entirely, rendering the need for effective approaches to alleviate this performance degradation, specifically in Mamba-based models. On the other hand, plain Mamba models do not experience performance degradation in processing textual data, as shown from the experiment results in the original Mamba papers [7, 8].
>
> | Model        | Mamba-790M | Mamba-1.4B | Mamba-2.8B | Mamba2-780M | Mamba2-1.3B | Mamba2-2.7B |
> |--------------|------------|------------|------------|-------------|-------------|-------------|
> | Size (M)     | 790        | 1,400      | 2,800      | 780         | 1,300       | 2,700       |
> | HellaSwag    | 55.1       | 59.1       | 66.1       | 54.9        | 59.9        | 66.6        |
> | LAMBADA      | 62.7       | 64.9       | 69.2       | 61.7        | 65.7        | 69.7        |
>
>
> Thus, we observe an imbalance of the processing capabilities of Mamba on visual and textual inputs, when scaled up in billion-parameter models. This phenomenon renders the need for more effective visual processing in large-scaled Mamba models to ensure that visual and textual features are adequately aligned.

---

> ### Author Response · Authors · 2024-11-21
> **Q1. Continued.**
>
> **Visual and textual latents in transformer-based models.** We argue that the issue of significant degradation in performance in visual task when scaled-up is unique to Mamba-based models, which requires more immediate remedies compared to transformer-based approaches to enable the full potential of the adaptation of Mamba models to MLLMs. Transformer models do not experience this scaling issue in vision or language tasks. However, given that typical MLLM approaches do not constrain visual inputs in any form, an autoregressive loss in visual tokens still demonstrate performance improvements in the EMU series and other works [9, 10, 11, 12]. Nevertheless, these approaches utilize complicated decoder structures that require additional training stage and data to effectively achieve visual self-supervision of LLMs. On the other hand, we show that a simple Mamba-based decoder is effective in achieving visual supervision, enabling our model to grasp more intricate visual details, which results in less visual hallucinations and better overall performance.

---

> ### Author Response · Authors · 2024-11-21
> **Q2. Could you provide a more objective, quantitative analysis of the loss of fine-grained visual cues in the LLM, as the current visualizations with just a few images seem insufficient? Is this phenomenon common across MLLMs, or is it specific to the use of the Mamba LLM?**
>
> **Q2. Could you provide a more objective, quantitative analysis of the loss of fine-grained visual cues in the LLM, as the current visualizations with just a few images seem insufficient? Is this phenomenon common across MLLMs, or is it specific to the use of the Mamba LLM?**
>
> **Motivation behind loss of fine-grained visual cues.** We discovered the problem of the loss of fine-grained visual cues when directly extracting and comparing intermediate features of the Cobra [13] model. Specifically, we find that visual features in deeper layers of the Mamba LLM exhibit increased portions of noise (manifested in unnatural specs in the image) and loss of structural integrity (manifested in the loss of topological structures) in the Cobra baseline model, which can be found in our visualizations in the main paper. Subsequently, we designed the hierarchical alignment loss to enhance the participation of intermediate visual features to alleviate the gradual loss of visual information.
>
> **Quantitative metrics to account for loss of fine-grained visual cues.** When composing this paper, we also attempted different ways to directly quantitatively visualize loss of fine-grained visual features, such as calculating the L2 distance or KL-divergence of intermediate features. However, these results do not reflect the actuality of loss of fine-grained visual cues, as different intermediate layers may in fact be responsible for different scales of visual intricacy, and simple metrics do not reflect how the model loses fine-grained information. We propose that, since we are dealing with intermediate features, it would require a model-specific metric to evaluate this loss of information, as different models would also possess different intermediate representations for the same sample input. Thus, it becomes extremely hard to visualize quantitatively the effect on fine-grained visual cues besides qualitatively displaying these features. We have also researched MLLM literature that tackles relevant subjects such as visual feature extraction [6, 14, 15, 16], visual feature fusion [17, 18] and did not find any suitable metrics. Consequently, we have resorted to hallucination benchmarks such as POPE and HallusionBench to evaluate the capability of models to effectively utilize fine-grained visual cues.

---

> ### Author Response · Authors · 2024-11-21
> **Q3. Why do the current results show that the Mamba LLM still lags behind the transformer architecture? In this case, does the speed advantage gained by using the Mamba architecture sufficiently compensate for the performance loss?**
>
> **Q3. Why do the current results show that the Mamba LLM still lags behind the transformer architecture? In this case, does the speed advantage gained by using the Mamba architecture sufficiently compensate for the performance loss?**
>
> In short, we believe that Mamba-based LLMs do not lag behind the transformer architecture but rather show competitive performances with them, depending on the training procedure. We believe that the enhanced performance of some transformer-based models ultimately lies in the use of more and better-quality data. Furthermore, all Mamba-based models are at a disadvantage (in terms of training data) when compared with transformer-based models, and a completely fair comparison between them in the current stage is unavailable. I will divide my answer to this question into three separate segments: 1. Analysis of current quantitative results, 2. Analysis of transformer-based methods, and 3. Analysis of Mamba-based methods.
>
>
> **Quantitative analysis of current results.** We believe that Mamba-based models offer competitive performances to the current suite of small-scaled transformer models. For instance, in Table 1 of the original paper, both versions of our model surpass the performance of LLaVA-Phi and MobileVLM on all metrics, except the MMB metric for EMMA-V2. Our performance is a bit weaker than MobileVLM v2, but we believe that this performance difference should be expected due to the use of different training datasets. We also surpass TinyLLaVA on all but the VQA-v2 and MMB metric.
>
> **Analysis of transformer-based methods.** 1. General facts about transformer-based LLMs. All transformer methods focus on two-stage training paradigm which consists of a pretraining phase and a finetuning phase. All backbone LLM was trained on 1.3T (MobileLLaMA) and 1.4T (Phi-2-2.7B) textural tokens prior to the MLLM pretrain and finetuning stage. 2. MobileVLM2. In this section, we focus on addressing the performance differences to MobileVLM v2, as opposed to the original MobileVLM. Architecturally, the improvement of MobileVLM to MobileVLM V2 is on the lightweight downsample projector, which replaces point-wise and depth-wise convolutions with average pooling and addition, mainly for performance gains. We suggest that the performance gain between these two models most likely lies in the amount of data used to pre-train and fine-tune both models. For MobileVLM, they utilize the same training corpora as LLaVA v-1.5, which consists of 1. the 558K subset of the LAION-CC-SBU dataset for pretraining and 2. the mixture of 665k dataset for finetuning, which includes samples from COCO, GQA, OCR-VQA, TextVQA, and Visual Genome. The total is around 1.2M data for the entire training procedure. For MobileVLM V2, they utilize a mixture of 3.6M data. This includes 1. 1.2M ShareGPT4V-PT data for pretraining, 2. A mixture of 2.4M data for finetuning, which includes samples from Visual Dialogue, TextVQA, VSR, VIGC, IConQA, SQA, COCO, SBU, and ShareGPT4V. As shown, MobileVLM V2 utilizes three times the amount of data, which may explain its significant performance gains over the original MobileVLM paper.
>
>
> **Analysis of Mamba-based methods.** 1. General facts about Mamba-based LLMs. Both Cobra and EMMA utilize a single finetuning phase, as the inclusion of pretraining phase does not significantly affect model performance. To conduct a fair comparison with our baseline, we also utilize the same dataset as Cobra to conduct our experiments. Furthermore, all Mamba-based backbones, unlike their transformer counterparts, are trained on the SlimPj [19] (MambaV1) and Pile [20] dataset (MambaV2), with only 627B and 300B tokens respectively. Thus, we believe it is impossible to make a completely fair comparison between transformer and Mamba-based models unless the underlying LLM as well as the entire process of multi-modal adaptation of these models utilize the same exact set of data. However, due to limited computational power, we do not have the capability of training a Mamba LLM from scratch, and thus the completely fair comparison between these methods remain unavailable. Nevertheless, Mamba-based models are able to achieve comparable performances with transformer-based models, despite having overall less training data and less training stages.

---

> ### Author Response · Authors · 2024-11-21
> **Q3. Continued.**
>
> **Speed vs. Performance trade-offs.** In the current case where Mamba models experience worse performance compared to some transformer models, we believe the use of either model depends on the deployment case. For instance, given the 3-times faster inference speed of Mamba models, they are certainly more effective to be deployed in scenarios that require rapid responses, such as real-time translation, fraud detection and edge-device maneuvering. However, for platforms that are less time-demanding, transformer models may still be the solution given their reliability and performance gains. Nevertheless, our research aims to more effectively adapt Mamba-based models for application within the domain of MLLMs, thereby providing impetus for future investigations aimed at fully harnessing the latent capabilities inherent in the Mamba architecture.

---

> ### Author Response · Authors · 2024-11-21
> **Q4. Currently, models use ViT as the image encoder. Is it possible to build an MLLM entirely based on Mamba? What potential advantages and challenges might this approach entail?**
>
> **Q4. Currently, models use ViT as the image encoder. Is it possible to build an MLLM entirely based on Mamba? What potential advantages and challenges might this approach entail?**
>
>
> **Fully Mamba-based MLLM.** The potential advantage of a fully Mamba-based model is the further improvement in inference speed given the Mamba architecture. While we believe that it is technically possible to build an MLLM entirely based on Mamba, this is not achievable given the currently available models, due to the lack of large pretrained visual Mamba models. As you may know, the typical framework of multimodal LLMs consists of a pretrained visual encoder, a multimodal projector, and a pretrained backbone LLM. However, the existence of a pre-trained large visual Mamba remains unavailable. In fact, the largest off-the-shelf pretrained visual Mamba models [4, 5] are less than 100M in parameters. It is possible to directly replace the visual encoder and projector with Mamba layers, however, jointly training a newly-initialized vision encoder with the LLM backbone results in terrible performances across all metrics. As a preliminary study for this paper, we have trained a version of Cobra with the vision encoder replaced by a newly initialized VMamba encoder (around 700M in parameter). Evaluation on the GQA benchmark demonstrated a 35\% drop in performance, suggesting that a pretrained Mamba-based visual encoder is necessary for the adaption of a fully-Mamba based MLLM.

---

> ### Author Response · Authors · 2024-11-21
> **Q5. In transformer-based LLMs, is there a similar issue with pixel-level alignment? Would the method proposed in the paper also be applicable to transformer architectures?**
>
> **Q5. In transformer-based LLMs, is there a similar issue with pixel-level alignment? Would the method proposed in the paper also be applicable to transformer architectures?**
>
> Please see our reply to Q1.

---

> ### Author Response · Authors · 2024-11-21
> **References.**
>
> 1. Alexey Dosovitskiy. An image is worth 16x16 words: Transformers for image recognition at scale. arXiv preprint arXiv:2010.11929, 2020.
> 2. Alex Krizhevsky, Ilya Sutskever, and Geoffrey E Hinton. Imagenet classification with deep convolutional neural networks. Advances in neural information processing systems, 25, 2012.
> 3. Yi Tay, Mostafa Dehghani, Samira Abnar, Yikang Shen, Dara Bahri, Philip Pham, Jinfeng Rao, Liu Yang, Sebastian Ruder, and Donald Metzler. Long range arena: A benchmark for efficient transformers. arXiv preprint arXiv:2011.04006, 2020.
> 4. Yue Liu, Yunjie Tian, Yuzhong Zhao, Hongtian Yu, Lingxi Xie, Yaowei Wang, Qixiang Ye, and Yunfan Liu. Vmamba: Visual state space model. arXiv preprint arXiv:2401.10166, 2024.
> 5. Lianghui Zhu, Bencheng Liao, Qian Zhang, Xinlong Wang, Wenyu Liu, and Xinggang Wang. Vision mamba: Efficient visual representation learning with bidirectional state space model. arXiv preprint arXiv:2401.09417, 2024.
> 6. Sucheng Ren, Xianhang Li, Haoqin Tu, Feng Wang, Fangxun Shu, Lei Zhang, Jieru Mei, Linjie Yang, Peng Wang, Heng Wang, et al. Autoregressive pretraining with mamba in vision. arXiv preprint arXiv:2406.07537, 2024.
> 7. Albert Gu and Tri Dao. Mamba: Linear-time sequence modeling with selective state spaces. arXiv preprint arXiv:2312.00752, 2023.
> 8. Tri Dao and Albert Gu. Transformers are ssms: Generalized models and efficient algorithms through structured state space duality. arXiv preprint arXiv:2405.21060, 2024.
> 9. Quan Sun, Qiying Yu, Yufeng Cui, Fan Zhang, Xiaosong Zhang, Yueze Wang, Hongcheng Gao, Jingjing Liu, Tiejun Huang, and Xinlong Wang. Emu: Generative pretraining in multimodality. In The Twelfth International Conference on Learning Representations, 2023a.
> 10. Quan Sun, Qiying Yu, Yufeng Cui, Fan Zhang, Xiaosong Zhang, Yueze Wang, Hongcheng Gao, Jingjing Liu, Tiejun Huang, and Xinlong Wang. Generative pretraining in multimodality. arXiv preprint arXiv:2307.05222, 2023b.
> 11. Quan Sun, Yufeng Cui, Xiaosong Zhang, Fan Zhang, Qiying Yu, Yueze Wang, Yongming Rao, Jingjing Liu, Tiejun Huang, and Xinlong Wang. Generative multimodal models are in-context learners. In Proceedings of the IEEE/CVF Conference on Computer Vision and Pattern Recognition, pp. 14398–14409, 2024.
> 12. Fei Zhao, Taotian Pang, Chunhui Li, Zhen Wu, Junjie Guo, Shangyu Xing, and Xinyu Dai. Aligngpt: Multi-modal large language models with adaptive alignment capability. arXiv preprint arXiv:2405.14129, 2024.
> 13. Han Zhao, Min Zhang, Wei Zhao, Pengxiang Ding, Siteng Huang, and Donglin Wang. Cobra: Extending mamba to multi-modal large language model for efficient inference. arXiv preprint arXiv:2403.14520, 2024a.
> 14. Guanqun Wang, Xinyu Wei, Jiaming Liu, Ray Zhang, Yichi Zhang, Kevin Zhang, Maurice Chong, and Shanghang Zhang. Mr-mllm: Mutual reinforcement of multimodal comprehension and vision perception. arXiv preprint arXiv:2406.15768, 2024.
> 15. Alaaeldin El-Nouby, Michal Klein, Shuangfei Zhai, Miguel Angel Bautista, Alexander Toshev, Vaishaal Shankar, Joshua M Susskind, and Armand Joulin. Scalable pre-training of large autoregressive image models. arXiv preprint arXiv:2401.08541, 2024.
> 16. Bo Li, Peiyuan Zhang, Jingkang Yang, Yuanhan Zhang, Fanyi Pu, and Ziwei Liu. Otterhd: A high-resolution multi-modality model. arXiv preprint arXiv:2311.04219, 2023.
> 17. Xiangyu Zhao, Xiangtai Li, Haodong Duan, Haian Huang, Yining Li, Kai Chen, and Hua Yang. Mg-llava: Towards multi-granularity visual instruction tuning. arXiv preprint arXiv:2406.17770, 2024b
> 18. Zonghao Guo, Ruyi Xu, Yuan Yao, Junbo Cui, Zanlin Ni, Chunjiang Ge, Tat-Seng Chua, Zhiyuan Liu, and Gao Huang. Llava-uhd: an lmm perceiving any aspect ratio and high-resolution images. In European Conference on Computer Vision, pp. 390–406. Springer, 2025.
> 19. Zhiqiang Shen, Tianhua Tao, Liqun Ma, Willie Neiswanger, Zhengzhong Liu, Hongyi Wang, Bowen Tan, Joel Hestness, Natalia Vassilieva, Daria Soboleva, et al. Slimpajama-dc: Understanding data combinations for llm training. arXiv preprint arXiv:2309.10818, 2023.
> 20. Leo Gao, Stella Biderman, Sid Black, Laurence Golding, Travis Hoppe, Charles Foster, Jason Phang, Horace He, Anish Thite, Noa Nabeshima, et al. The pile: An 800gb dataset of diverse text for language modeling. arXiv preprint arXiv:2101.00027, 2020.

---

> ### Author Response · Authors · 2024-11-24
> **Looking forward to further discussions!**
>
> Dear Reviewer T8Kc,
>
> Thank you for your insightful comments. We were wondering if our response and revision have resolved your concerns. We have attempted to address your initial questions through our replies and are eager to clarify any further points you might raise. Please feel free to provide additional feedback. We greatly appreciate your continued engagement.
>
> Best regards, Authors

---

> ### Comment · Area_Chair_5dM9 · 2024-11-26
>
> Dear Reviewer T8Kc,
>
> Could you kindly review the rebuttal thoroughly and let us know whether the authors have adequately addressed the issues raised or if you have any further questions.
>
> Best,
>
> AC

---

> ### Comment · Reviewer_T8Kc · 2024-12-03
>
> Thank you for your response. My concerns are mostly addressed. I would like to retain the initial score.

---

### Official Review · Reviewer_5nFQ · 2024-11-04

**Soundness:** 3
**Presentation:** 3
**Contribution:** 2
**Rating:** 6
**Confidence:** 4

**Summary:**

This paper proposes a Mamba based integration for vision-language models. Similar to the VLM based framework where there are one vision encoder, one projector, and one LLM. The proposed method takes one image and its corresponding caption as input, followed by multi-scale feature extraction and fusion. A pixel-wise alignment loss is applied to the visual features for structural visual cues preservation, and an autoregressive NLP loss is applied to the textural feature. Experiments are conducted on several benchmark datasets.

**Strengths:**

1. Pixel-wise alignment loss is interesting. The LLM outputs are text tokens and are further decoded into the image domain for measuring pixel-level similarity with the input image.
2. The hierarchical alignment via multi-scale fusion is developed to retain fine-grained visual features
3. The experimental results seem promising on the benchmark datasets.

**Weaknesses:**

1. While the main contribution is set for the loss design of model training, the proposed loss function seems rather general and not specifically for Mamba based structure. The general pixel-wise alignment loss seems functional for all VLMs, rather than only mamba structure. Can you clarify how the proposed loss function leverages or is tailored to the unique properties of Mamba architectures compared to transformer-based VLMs?
2. The inputs require both images and captions, how the captions are acquired are not illustrated well. A more detailed description of the data preparation process is desired here. Also, the decoded procedure via Mamba is not illustrated clearly. Is SFT or a similar finetuning process still needed for training VLM? Is related VQA data still required? Such technical details are not shown clearly in the manuscript to show the overall model tuning picture.  It is better to provide a step-by-step explanation of the decoding procedure and clarification on the fine-tuning process to help address the ambiguity.
3. The pixel-alignment loss is set for structure preservation, the role of LLM here is not clear as the caption has been sent to it. So LLM seems to function as a single mapping to fulfill this objective loss. Can you elaborate on the specific role of the LLM in the context of the pixel-alignment loss?
4. Feature fusion is not motivated well and seems a general operation for representation enhancement. It would be better to provide a more detailed justification for the choice of feature fusion technique. Specifically, an explanation is expected of how your approach differs from or improves upon standard feature fusion methods, particularly in the context of Mamba-based architectures.

**Questions:**

Overall, the pixel-alignment loss design is interesting but there are many ambiguious technical details to make the contribution clear. Further clarification is required to better position the proposed method within the scope of the visual instruction tuning design of VLMs

---

> ### Author Response · Authors · 2024-11-21
> **Q1. Can you clarify how the proposed loss function leverages or is tailored to the unique properties of Mamba architectures compared to transformer-based VLMs?**
>
> **Q1. Can you clarify how the proposed loss function leverages or is tailored to the unique properties of Mamba architectures compared to transformer-based VLMs?**
>
> To answer this question, we present more background information detailing the motivation of our work, and how our work differentiates itself from its precursor. Then, we answer the question of whether the proposed loss function leverages or is tailored to the unique properties of Mamba architectures.
>
> **Clarification of Motivation.** Our research is grounded in the observation that Mamba models exhibit an imbalance in handling visual and textual information. We argue for the imbalanced quality of visual and textual latents in Mamba models through two perspectives, architecturally and empirically.
>
>
> **Architectural imbalance.** The inherent linear scanning mechanism of Mamba poses challenges in effectively extracting visual features on the same level as textual features. On one hand, transformer-based [1] approaches utilize the self-attention mechanism that extracts global relationships between different parts of the input sequence. They also utilize a position encoding for learning spatial relationships of visual tokens. On the other hand, CNN-based approaches [2] utilize local receptive fields to connect each neuron to a local region of the input data, allowing interaction of neighboring pixels in all directions. The effectiveness of both methods in processing visual information lies in their ability to account for global, or to the least, local clusters of the visual input. However, Mamba utilizes a selective-scan mechanism that only processes sequences in a linear order. While Mamba models achieve excellent performances in textual data, effectively capturing long-term dependencies in multiple language benchmarks [3], Mamba models possess difficulties in handling image data, where pixel-wise relationships may span multiple rows and columns. Visual Mamba models have been proposed to counteract this issue through combining additional scans from different directions [4] and utilizing a similar position encoding as ViTs [5]. However, they nevertheless still utilize the linear selective-scan mechanism. In the case where features possess large embedding dimensions, which is typical in MLLM scenarios, this becomes hard for the model to extract visual cues that are far apart spatially, as the degree of freedom between each visual token across different rows and columns become huge.
>
>
>
>
> **Empirical imbalance.** Mamba models experiences an imbalance in the empirical evaluation of vision tasks and textual tasks, when scaled up in parameters. [6] observes the performance degradation of visual Mamba models when scaled up. Below is a table taken directly from their paper [7, 8].
>
> | Model        | VIM-B | VIM-L | VIM-H |
> |--------------|-------|-------|-------|
> | Size (M)     | 98    | 340   | 755   |
> | ImageNet Acc.| 81.2  | 81.0  | Collapsed |
>
>
> As shown, plain Mamba models experience a severe degradation in processing visual features when scaled up. In the 700M parameter-scale, the visual Mamba collapses entirely, rendering the need for effective approaches to alleviate this performance degradation, specifically in Mamba-based models. On the other hand, plain Mamba models do not experience performance degradation in processing textual data, as shown from the experiment results in the original Mamba papers [7, 8].
>
> | Model        | Mamba-790M | Mamba-1.4B | Mamba-2.8B | Mamba2-780M | Mamba2-1.3B | Mamba2-2.7B |
> |--------------|------------|------------|------------|-------------|-------------|-------------|
> | Size (M)     | 790        | 1,400      | 2,800      | 780         | 1,300       | 2,700       |
> | HellaSwag    | 55.1       | 59.1       | 66.1       | 54.9        | 59.9        | 66.6        |
> | LAMBADA      | 62.7       | 64.9       | 69.2       | 61.7        | 65.7        | 69.7        |
>
> Thus, we observe an imbalance of the processing capabilities of Mamba on visual and textual inputs, when scaled up in billion-parameter models. This phenomenon renders the need for more effective visual processing in large-scaled Mamba models to ensure that visual and textual features are adequately aligned.
>
> **Is pixel-wise alignment unique to Mamba?** The answer to this question is Yes and No, depending on different perspectives. We will answer this question in two different perspectives to showcase the meaningfulness of our research in the grand schema of MLLMs.

---

> ### Author Response · Authors · 2024-11-21
> **Q1. Continued.**
>
> **Pixel-wise alignment is unique to Mamba.** EMMA is unique in the sense that it is the first to consider multimodal alignment specifically in Mamba-based MLLMs. We first observe the inconsistent performance of Mamba models with respect to visual and textual inputs when scaled-up in parameters. In specific, Mamba models tend to experience a significant performance decline in processing visual data when scaled-up, despite them being able to seamlessly handle textual data in large-scaled settings. On the other hand, to deploy Mamba models in MLLM settings, where large-scaled Mamba models are necessary in processing both textual and visual information, it is crucial to preserve a balance in the quality of visual and textual features to achieve multi-modal alignment. Previous Mamba MLLM works such as VL Mamba and Cobra directly replaces the transformer LLM with the Mamba LLM model without addressing this imbalance of feature quality. Furthermore, as a preliminary study for this work, we have also tried transformer-based multimodal alignment approaches that enhance visual features or encourage multimodal alignment, such as visual-feature compression [9], masked-alignment [10, 11], and contrastive losses [12, 13], all of which fail in the Mamba LLM setting. Here, we show the respective model performances on the TextVQA dataset, which we find is a good indicator for overall model performance.
>
>
> | Model        | EMMA-V1 | EMMA-V2 | Cobra | Compression | Masked | Contrastive |
> |--------------|---------|---------|-------|-------------|--------|-------------|
> | TextVQA      | 57.2    | 56.2    | 52.4  | 45.8        | 46.1   | 50.2        |
>
>
>
> As shown, transformer-based approaches tailored for multimodal alignment in MLLM settings may in fact negatively impact model performance, which renders the necessity to devise Mamba-specific multi-modal alignment approaches. On the other hand, autoregressive pretraining on the visual features are shown to improve visual Mamba models [6], and for the first time allows visual Mamba models to scale up to large-sized models (600M). Given the effectiveness of autoregressive pretraining in promoting the single-modal visual Mamba model, we extend this approach to the multimodal LLM setting. Subsequently, we propose the pixel-wise alignment loss to encourage the generated autoregressive features to structurally align with the input image. Furthermore, when visualizing the LLM features, we notice a gradual decline of the intermediate visual features, causing the final generated visual feature to degrade. In response, we propose the multi-scale feature fusion module to tackle this issue, leading to better preservation of intermediate visual features, inducing better multimodal alignment of visual and textual features. In essence, our method specifically tackles the issue of multi-modal alignment in Mamba models, while the direct application of transformer-based multimodal methods have failed. As a result, EMMA demonstrates improved performances on a variety of metrics, and most importantly, significantly reduces the degree of visual hallucinations in the HallusionBench benchmark, surpassing transformer models with twice the size.
>
> **Pixel-wise alignment is not unique to Mamba?**
> As presented in our work, transformer-based models have also utilized similar methodologies [14, 15] in autoregressively generating visual features along with textual cues. Thus, the general idea of image-wise autoregressive loss is not unique to Mamba, in the sense that transformer-based methods also benefit from it. However, we still believe our implementation and execution of this idea is unique and meaningful in the setting of MLLMs, specifically in the sense of efficiency. While the aforementioned transformer methods achieve overall improvements in various MLLM benchmarks, they nevertheless introduce additional training stages that are specifically tailored for the visual decoder, which is used to generate the autoregressively generated visual images. Their decoders are also complex, usually consisting of stable-diffusion models that yield huge amounts of parameters. On the other hand, our Mamba-based visual decoder is extremely light-weight (around 200M parameters), and trained end-to-end together with the textual token loss. Thus, it requires no extra training data or training stages. Hence, we show an efficiently implemented Mamba-based decoder is sufficient to achieve image-wise self-supervision to the multimodal LLM.

---

> ### Author Response · Authors · 2024-11-21
> **Q2. The inputs require both images and captions, how the captions are acquired are not illustrated well. A more detailed description of the data preparation process is desired here.**
>
> **Q2. The inputs require both images and captions, how the captions are acquired are not illustrated well. A more detailed description of the data preparation process is desired here.**
>
> **Training Data and Preparation.** We train our model using the combination of LLaVA-v1.5-mixed-665k, LVIS-Instruct-4V, and LRV-Instruct datasets. Here we will provide a more detailed description of the composition of each dataset, as well as the overall data preparation process.
>
> **LLaVA-v1.5-mixed-665k.** This is the same dataset used for finetuning the LLaVA-v1.5 model [16]. We provide the breakdown for this dataset as a comma-separated list with the origin and corresponding number of samples: LLaVA 158K, ShareGPT 40K, VQAv2 83K, GQA 72K, OKVQA 9K, OCRVQA 80K, A-OKVQA 66K, TextCaps 22K, RefCOCO 48K, and VG 86K, for a total of 665K data samples. According to the LLaVA-v1.5 paper, these data are preprocessed as follows: 1. For all VQA datasets, QA pairs from the same training image are merged into a single conversation. 2. For ShareGPT, they filter out invalid conversations. They also truncate long conversations that surpass 2048 tokens. 3. Each QA pair in A-OKVQA is augmented k times, where k is the number of choices per question, to counterbalance the lack of multiple-choice data. 4. They sample 80K conversations from OCRVQA. 5. For Visual Genome, they sample 10 annotations for images with additional annotations. 6. For RefCOCO, conversations are dissected into segments, each containing fewer than 10 conversations. 7. All data splits are concatenated together to form the final 665K dataset.
>
>
> **LVIS-Instruct-4V.** This is the dataset presented by [17], in an attempt to curate a fine-grained instruction-following dataset by explicitly conditioning the generation of instruction data on both language and visual inputs. To generate this dataset, they leverage image data from LVIS [18], as well as their fine-grained annotations to prompt the GPT-4V model to undertake two key tasks: (1) generate conversational question-answer lists through self-reasoning and (2) produce high-quality image descriptions guided by precise bounding box information. In the end, they use 110K images from LVIS and generate 220K high-quality visual instructions, which consist of 110K conversational data and 110K descriptional data. These images along with GPT-4V-generated captions form this dataset.
>
> **LRV-Instruct.** This dataset [19] leverages GPT4 to cover open-ended positive and negative instructions in different linguistic styles for images in the Visual Genome dataset. They use GPT4 to create instruction-following data with the image size, bounding boxes, and dense captions as input, and generate caption instances in both declarative and interrogative formats. Afterwards, they remove instances with answers longer than 30 words and those with unneeded content. This results a total of over 400k image-visual instruction pairs after filtering.
>
>
> **Model Preprocessing.** After combining these three datasets, we finetune our entire model on it. Because we utilize pre-trained vision encoders and Mamba LLM, for data pre-processing, we also adopt the same tokenizer and image transforms from the respective components. For Mamba LLM, we utilize the GPTNeoXTokenizerFast tokenizer to convert input text into textual tokens. For both the SigLIP and DinoV2 ViTs, we use their respective image transforms, as listed below: 1. SigLIP: Resize to (384, 384), center crop, normalize by mean 0.5 and std 0.5. 2. DinoV2: Resize to (384, 384), center crop, normalize by mean 0.485 and std 0.225. We will add such details into our supplementary.

---

> ### Author Response · Authors · 2024-11-21
> **Q3. Also, the decoded procedure via Mamba is not illustrated clearly.**
>
> **Q3. Also, the decoded procedure via Mamba is not illustrated clearly. Is SFT or a similar finetuning process still needed for training VLM? Is related VQA data still required? Such technical details are not shown clearly in the manuscript to show the overall model tuning picture. It is better to provide a step-by-step explanation of the decoding procedure and clarification on the fine-tuning process to help address the ambiguity.**
>
> **Decoder procedure.** We do not include a separate stage for training the decoder, but rather jointly optimize the decoder, LLM, and projector in a singular finetuning stage. Thus, the overall model tuning picture consists of a single finetuning stage where we only freeze the vision encoder, and jointly tune the MLP projector, LLM, and decoder on all available training data, for 2 epochs. During training, the model concurrently generates a textual response used for the textual autoregressive loss, and a visual response used for the pixel-alignment loss. Both losses are combined and optimized simultaneously. The decoder does not use extra VQA data to be optimized.

---

> ### Author Response · Authors · 2024-11-21
> **Q4. Can you elaborate on the specific role of the LLM in the context of the pixel-alignment loss?**
>
> **Q4. The pixel-alignment loss is set for structure preservation, the role of LLM here is not clear as the caption has been sent to it. So LLM seems to function as a single mapping to fulfill this objective loss. Can you elaborate on the specific role of the LLM in the context of the pixel-alignment loss?**
>
> In short, we want our pixel-wise alignment loss to directly enforce the LLM to focus not only on the generation of the textual response but also on learning better quality visual representations and preserving relevant visual information for multimodal alignment. As mentioned in the response to Q1, Mamba models are inherently weaker in processing visual features, especially in larger-scale models. In the context of LLMs, where parameters are in the scope of billions, we hypothesize that this ineffectiveness in processing visual information may severely impact the ability for Mamba models to align visual and textual features, necessitating additional supervision of visual information in Mamba MLLMs.
>
>
> **The role of LLMs in conventional MLLMs.** Conventional MLLMs usually consists of a ViT, an MLP projector, and an LLM. The entire training process of MLLMs endows the LLM to accept not only textual information, but also visual information through the vision encoder, to produce textual responses based on multimodal input. Thus, the LLM here functions as a mapping from image space and text space to text space, optimized solely through a loss function that constrains the text space response.
>
> **Imbalance between visual and textual features.** We observe an inherent flaw in the optimization of current MLLMs. Despite handling both visual and textual data, the visual features within the network does not seem to be conditioned in any form. Combined with the observation of performance degradation of large-scale visual Mamba models, we extend the autoregressive loss from the text space to image space as well, to enable the linguistic capabilities of the LLM while preserving structural visual cues of the image, ultimately enabling better multi-modal alignment between the image and text modality. During training, the LLM here functions as a mapping from image space and text space to image space and text space, optimized through a loss function that constrains both textual responses and the structure of visual information. Thus, the LLM is trained to simultaneously generate a logical response to the input image and text data, while also able to preserve structural visual information, reducing visual hallucinations. During testing, the visual constraints are unnecessary, and discarded entirely.

---

> ### Author Response · Authors · 2024-11-21
> **Q5. Feature fusion is not motivated well.**
>
> **Q5. Feature fusion is not motivated well and seems a general operation for representation enhancement. It would be better to provide a more detailed justification for the choice of feature fusion technique. Specifically, an explanation is expected of how your approach differs from or improves upon standard feature fusion methods, particularly in the context of Mamba-based architectures.**
>
>
> **Mamba-based feature-fusion methods.** To our knowledge, current Mamba-based feature fusion methods include MambaDFuse [20], fusion-Mamba [21] and FusionMamba [22]. MambaDFuse proposes a dual-phase feature fusion module that consists of shallow fuse and deep fuse modules. The shallow fuse module achieves lightweight exchange of features from multiple modalities, while the deep fuse module, a newly designed Multi-modal Mamba (M3) block, guides the generation of modality-fused features and incorporates local detail characteristics from different modalities. Fusion-Mamba proposes a Fusion-Mamba block (FMB) to map cross-modal features into a hidden state space for interaction, thereby reducing disparities between cross-modal features and enhancing the representation consistency of fused features. FusionMamba designs a Mamba model suitable for fusion tasks by integrating the Visual State Space Model with dynamic convolution and channel-wise attention.
>
>
> **Improvements from previous Mamba-based methods.** We believe the novelty of our work lies in two perspectives. First, we are the first to examine feature-fusion of Mamba models in the context of multimodal LLMs. Previous feature fusion works only focus on multi-stream multimodal models which do not utilize a Mamba-based LLM, but rather consist of separate branches with similar sizes to encode both modalities. Thus, the differences in underlying model architectures renders different design choices for feature fusion methods. Previous Mamba-based multimodal LLM methods, on the other hand, adapts the Mamba LLM to typical MLLM frameworks without considering feature fusion. Our work is the first that considers feature fusion in MLLMs to boost the quality of visual features to ensure better multimodal alignment, justified with noticeable performance gains empirically. Secondly, we are the first to examine feature-fusion of same modality features in Mamba-based models. Previous feature fusion works only consider the fusion of multimodal features, rather than augmenting useful information of same modality features from different scales. Our work proposes a novel Mamba-based multi-scale feature fusion module that effectively combines visual features from different layers, alleviating the gradual loss of visual information in the Mamba LLM.

---

> ### Author Response · Authors · 2024-11-21
> **References**
>
> 1. Alexey Dosovitskiy. An image is worth 16x16 words: Transformers for image recognition at scale. arXiv preprint arXiv:2010.11929, 2020.
> 2. Alex Krizhevsky, Ilya Sutskever, and Geoffrey E Hinton. Imagenet classification with deep convolutional neural networks. Advances in neural information processing systems, 25, 2012.
> 3. Yi Tay, Mostafa Dehghani, Samira Abnar, Yikang Shen, Dara Bahri, Philip Pham, Jinfeng Rao, Liu Yang, Sebastian Ruder, and Donald Metzler. Long range arena: A benchmark for efficient transformers. arXiv preprint arXiv:2011.04006, 2020.
> 4. Yue Liu, Yunjie Tian, Yuzhong Zhao, Hongtian Yu, Lingxi Xie, Yaowei Wang, Qixiang Ye, and Yunfan Liu. Vmamba: Visual state space model. arXiv preprint arXiv:2401.10166, 2024.
> 5. Lianghui Zhu, Bencheng Liao, Qian Zhang, Xinlong Wang, Wenyu Liu, and Xinggang Wang. Vision mamba: Efficient visual representation learning with bidirectional state space model. arXiv preprint arXiv:2401.09417, 2024.
> 6. Sucheng Ren, Xianhang Li, Haoqin Tu, Feng Wang, Fangxun Shu, Lei Zhang, Jieru Mei, Linjie Yang, Peng Wang, Heng Wang, et al. Autoregressive pretraining with mamba in vision. arXiv preprint arXiv:2406.07537, 2024.
> 7. Albert Gu and Tri Dao. Mamba: Linear-time sequence modeling with selective state spaces. arXiv preprint arXiv:2312.00752, 2023.
> 8. Tri Dao and Albert Gu. Transformers are ssms: Generalized models and efficient algorithms through structured state space duality. arXiv preprint arXiv:2405.21060, 2024.
> 9. Wentong Li, Yuqian Yuan, Jian Liu, Dongqi Tang, Song Wang, Jie Qin, Jianke Zhu, and Lei Zhang. Tokenpacker: Efficient visual projector for multimodal llm. arXiv preprint arXiv:2407.02392, 2024.
> 10. David Mizrahi, Roman Bachmann, Oguzhan Kar, Teresa Yeo, Mingfei Gao, Afshin Dehghan, and Amir Zamir. 4m: Massively multimodal masked modeling. Advances in Neural Information Processing Systems, 36, 2024.
> 11. Yonatan Bitton, Gabriel Stanovsky, Michael Elhadad, and Roy Schwartz. Data efficient masked language modeling for vision and language. arXiv preprint arXiv:2109.02040, 2021.
> 12. Sheng Shen, Liunian Harold Li, Hao Tan, Mohit Bansal, Anna Rohrbach, Kai-Wei Chang, Zhewei Yao, and Kurt Keutzer. How much can clip benefit vision-and-language tasks? arXiv preprint arXiv:2107.06383, 2021.
> 13. Alec Radford, Jong Wook Kim, Chris Hallacy, Aditya Ramesh, Gabriel Goh, Sandhini Agarwal, Girish Sastry, Amanda Askell, Pamela Mishkin, Jack Clark, et al. Learning transferable visual models from natural language supervision. In International conference on machine learning, pp.8748–8763. PMLR, 2021
> 14. Quan Sun, Qiying Yu, Yufeng Cui, Fan Zhang, Xiaosong Zhang, Yueze Wang, Hongcheng Gao, Jingjing Liu, Tiejun Huang, and Xinlong Wang. Emu: Generative pretraining in multimodality. In The Twelfth International Conference on Learning Representations, 2023a.
> 15. Quan Sun, Qiying Yu, Yufeng Cui, Fan Zhang, Xiaosong Zhang, Yueze Wang, Hongcheng Gao, Jingjing Liu, Tiejun Huang, and Xinlong Wang. Generative pretraining in multimodality. arXiv preprint arXiv:2307.05222, 2023b.
> 16. Haotian Liu, Chunyuan Li, Yuheng Li, and Yong Jae Lee. Improved baselines with visual instruction tuning. In Proceedings of the IEEE/CVF Conference on Computer Vision and Pattern Recognition, pp. 26296–26306, 2024.
> 17. Junke Wang, Lingchen Meng, Zejia Weng, Bo He, Zuxuan Wu, and Yu-Gang Jiang. To see is to believe: Prompting gpt-4v for better visual instruction tuning. arXiv preprint arXiv:2311.07574, 2023.
> 18. Agrim Gupta, Piotr Dollar, and Ross Girshick. Lvis: A dataset for large vocabulary instance segmentation. In Proceedings of the IEEE/CVF conference on computer vision and pattern recognition, pp. 5356–5364, 2019.
> 19. Fuxiao Liu, Kevin Lin, Linjie Li, Jianfeng Wang, Yaser Yacoob, and Lijuan Wang. Mitigating hallucination in large multi-modal models via robust instruction tuning. In The Twelfth International Conference on Learning Representations, 2023.
> 20. Zhe Li, Haiwei Pan, Kejia Zhang, Yuhua Wang, and Fengming Yu. Mambadfuse: A mamba-based dual-phase model for multi-modality image fusion. arXiv preprint arXiv:2404.08406, 2024.
> 21. Wenhao Dong, Haodong Zhu, Shaohui Lin, Xiaoyan Luo, Yunhang Shen, Xuhui Liu, Juan Zhang, Guodong Guo, and Baochang Zhang. Fusion-mamba for cross-modality object detection. arXiv preprint arXiv:2404.09146, 2024.
> 22. Xinyu Xie, Yawen Cui, Chio-In Ieong, Tao Tan, Xiaozhi Zhang, Xubin Zheng, and Zitong Yu. Fusionmamba: Dynamic feature enhancement for multimodal image fusion with mamba. arXiv preprint arXiv:2404.09498, 2024.

---

> ### Author Response · Authors · 2024-11-24
> **Looking forward to further discussions!**
>
> Dear Reviewer 5nFQ,
>
> Thank you for your insightful comments. We were wondering if our response and revision have resolved your concerns. We have attempted to address your initial questions through our replies and are eager to clarify any further points you might raise. Please feel free to provide additional feedback. We greatly appreciate your continued engagement.
>
> Best regards, Authors

---

> > ### Comment · Reviewer_5nFQ · 2024-11-25
> >
> > Thanks for the clarification, and most of my concerns have been resolved. I'd suggest the authors to incorporate more motivations/rationales in their camera-ready version if the paper got accepted.

---

> > > ### Author Response · Authors · 2024-11-25
> > > **Thank you again for you comments!**
> > >
> > > We greatly appreciate your feedback and up-rating of our paper. We will incorporate the motivation and rationales in the camera-ready version of the paper if accepted. Thank you again for your valuable time and input!

---

### Official Review · Reviewer_yCEA · 2024-11-05

**Soundness:** 2
**Presentation:** 2
**Contribution:** 2
**Rating:** 6
**Confidence:** 3

**Summary:**

Mamba has seen widespread use in large language models (LLMs) due to its exceptional efficiency. However, Mamba struggles with processing visual signals, limiting its application in multi-modal LLMs. To address this issue, this paper proposes improving Mamba’s ability to handle visual signals through structural and hierarchical alignment. The authors argue that Mamba’s difficulty with visual data stems from the lack of positional encoding, which causes the gradual loss of fine-grained spatial information during processing. To mitigate this, the paper introduces the EMMA method, which enhances the structural consistency between Mamba’s output visual features and the original image by reconstructing the image through a decoder. Additionally, EMMA incorporates cross-attention within Mamba, combining multi-level intermediate features via cross-attention to form the final output, thereby preventing the loss of fine-grained details in deeper layers. The results show that EMMA leads to improved performance.

**Strengths:**

- The proposed method is simple yet effective.

**Weaknesses:**

1. **Increased Complexity Due to Cross-Attention**:
   EMMA introduces cross-attention operations into the Mamba network, reverting the model’s original sub-quadratic complexity back to quadratic complexity, which undermines the original intent of using Mamba. Could the authors provide the computational complexity of the cross-attention operations compared to the overall method? Is cross-attention necessary for fusing intermediate features? Could simpler alternatives be explored to achieve the same goal?

2. **Training Costs and Fair Comparisons**:
   EMMA involves additional training steps, which should be explicitly detailed in the paper. Furthermore, when comparing EMMA to other methods, the additional training costs should be listed separately to ensure a fair comparison (training time, GPU hours and memory requirement). Since EMMA alters the original MLLM architecture, its inference complexity will also increase. Table 4 should include a breakdown of the increased complexity due to hierarchical alignment. For fairness, Table 4 should also provide EMMA’s inference speed when using Mamba LLM-2.8B.

3. **Limited Technical Contribution**:
   EMMA’s approach is somewhat simplistic, as it merely enhances the consistency between deep and shallow features and the original image at the output stage. This method requires extra training, substantial computational resources, and introduces additional structures during deployment. As a result, the technical contribution appears somewhat trivial. It would be better if this paper could provide more detailed analysis of how their method addresses specific challenges in multimodal learning that previous methods have struggled with.

**Questions:**

Please refer to the weaknesses.

---

> ### Author Response · Authors · 2024-11-21
> **Q1. Could the authors provide the computational complexity of the cross-attention operations compared to the overall method?**
>
> **Q1. Could the authors provide the computational complexity of the cross-attention operations compared to the overall method?**
>
> To summarize, cross-attention only introduces negligible amounts of parameters compared to the overall method, and thus do not contribute much to the overall complexity of our method.
>
> **Computation cost analysis.** We provide the following table for parameter counts of different components in our model. $Time_{Trn}$ and $Memory_{Trn}$ denote training-wise time and memory requirements, while $Time_{Eval}$ and $Memory_{Eval}$ denote evaluation-wise time and memory requirements. Decoder refers to the decoder for generating visual features for pixel-wise alignment, while MFF refers to the multi-scale feature fusion module for the hierarchical alignment. $CA$ refers to the sum of all additional cross-attention modules introduced, to separately account for the computational complexity of cross-attention operations. Note that these parameters are also accounted in MFF. Memory is in terms of MiB, and time is in terms of seconds. The memory of each component is calculated by $$memory_{post-module} - memory_{pre-module}$$ The time of each component is calculated by $$time_{post-module} - time_{pre-module}$$ The training results include additional loss calculation and optimization passes, which introduce additional time and memory usage. Note that the inference time reported here multiplied by 256 does not give the actual model inference time as presented in Table 4 (in original paper), this is due to the incorporation of cached operations during inference (which are common in all MLLM models, including transformers), which only necessitate one pass through the ViT and reduced tokens for the LLM during the text generation phase. Here we report actual processing times instead of cached times for the sake of performance measure, but note that cached times are more suitable and realistic for deployment. The following breakdown is performed uniformly with a batch size of 1, evaluated on a NVIDIA A100 GPU.
>
> | Components           | Parameters | Time_{Trn} | Memory_{Trn} | Time_{Eval} | Memory_{Eval} |
> |----------------------|------------|------------|--------------|-------------|---------------|
> | Visual Encoder (ViT) | 731M       | 2e-2       | 3,600        | 2e-2        | 3,600         |
> | Projection MLP       | 47M        | 2e-3       | 498          | 4e-4        | 460           |
> | MambaV1-2.8B LLM     | 2.8B       | 7e-1       | 25,266       | 7e-2        | 11,140        |
> | MambaV2-2.7B LLM     | 2.7B       | 7e-1       | 26,210       | 6e-2        | 11,054        |
> | Decoder              | 174M       | 5e-1       | 1,204        | N/A         | N/A           |
> | MFF                  | 266M       | 6e-3       | 1,346        | N/A         | N/A           |
> | CA     | 105M | 1e-3  | 892          | N/A         | N/A           |
>
> As shown, cross-attention modules only account **for a very small proportion (roughly 3 percent) of the model's optimization parameters**. Additionally, **the cross-attention here is only quadratic in terms of the length of image patches and does not vary based on the length of text**, hence not truly quadratic in the strictest sense. Hence the memory and time usage of the cross attention module is extremely small. Also, since the decoder and MFF modules are not utilized during inference time, they pose no additional computation cost during testing.

---

> ### Author Response · Authors · 2024-11-21
> **Q2. Is cross-attention necessary for fusing intermediate features? Could simpler alternatives be explored to achieve the same goal?**
>
> **Q2. Is cross-attention necessary for fusing intermediate features? Could simpler alternatives be explored to achieve the same goal?**
>
> To summarize, we believe that cross-attention specifically is not necessary for fusing intermediate features, and simpler alternatives can be explored to achieve the same goal. However, given the small computational complexity compared to the overall model and the effectiveness of cross-attention in previous feature fusion works, we resort to the use of cross-attention to fuse intermediate features in our work. We do show that replacing the cross-attention with linear attention achieves similar performance gains.
>
> **Cross-attention background and overview.** The cross attention module [1] replaces key and value vectors in self-attention with inputs from different sequences, enabling the calculation of attention scores based on alternative information. Currently, cross-attention is commonly used in multimodal models for representation fusion, as evidenced by various methods from basic MLLM approaches like Flamingo [2] and EVLM [3] to advanced representation fusion MLLM methods such as LLAVA-UHD [4]. Cross-attention has demonstrated its capabilities in fusing both same-modal and multi-modal features in these works, and is the reason why we have also opted for cross-attention.
>
> **The use of attention-mechanisms in Mamba.** The use of the attention mechanism in Mamba-based architecture is not uncommon. The authors for the original Mamba [5] paper has provided a version of attention-augmented Mamba2 model in their follow-up work Mamba2 [6], which shows improved performances without introducing significant computational overhead. Furthermore, hybrid Mamba-transformer architectures such as Jamba [7,8], Zamba [9], and Griffin [10] have been proposed to combine the efficiency of Mamba and state-space models with the benefits of transformer-based attentions.
>
> **Cross-attention in EMMA.** We observe that current Mamba-based multimodal models exhibit a gradual loss of fine-grained visual cues in deeper layers of the MLLM. Thus, in an attempt to reduce this information loss, we propose to use a hierarchical alignment module to counteract this loss of information. The essence of using cross-attention in our work lies in merging information from multiple layers of representation, ensuring that information from intermediate layers is also involved in visual representation alignment to prevent loss of image information at intermediate layers. Therefore, cross-attention is not mandatory, but only serves as a means to merge layer-wise information. Thus, the replacement of it with more efficient and simpler attention mechanisms, or usage of different representation fusion modules can be explored. Nevertheless, due to the minimal number of parameters involved, there may not be a significant difference in computational efficiency, and hence not explored in this work.
>
> **Alternative Attention Mechanisms in EMMA.** As an example, we replace the cross-attention modules with lucidrains version [11] of linear attention [12,13] in the EMMA-V1 model. This enables linear-wise computation of attention, and achieves the following performances:
>
> | Model      | GQA  | VizWiz | VQA$^{T}$ | POPE | MME   | MMB  | HBench |
> |------------|------|--------|-----------|------|-------|------|--------|
> | EMMA-V1-CA | 60.5 | 52.1   | 57.2      | 88.0 | 1572.8| 53.2 | 51.0   |
> | EMMA-V1-LA | 60.2 | 52.1   | 57.7      | 88.1 | 1560.0| 52.5 | 46.9   |
>
> EMMA-V1-CA denotes the original proposed model with cross-attention, while EMMA-V1-LA denotes EMMA but with linear attention instead. EMMA-V1-LA performs similarly to EMMA-V1-CA on the majority of benchmarks, besides HallusionBench, which experiences a 4\% drop. This could be attributed to the trade-off of sub-quadratic complexity and overall performance of linear attention. Nevertheless, EMMA-V1-LA still outperforms the baseline on HallusionBench by 5\%.

---

> ### Author Response · Authors · 2024-11-21
> **Q3. EMMA involves additional training steps, which should be explicitly detailed in the paper.**
>
> **Q3. EMMA involves additional training steps, which should be explicitly detailed in the paper.**
>
> In short, EMMA **does not involve additional training steps** besides a **singular finetuning stage** that jointly tunes the projector, LLM and decoder.
>
> **Training procedure of MLLMs.** We would like to first clarify the training procedure of vision-language models (VLMs). Here, we are specifically referring to models inspired by the LLaVA [14] architecture, which consists of a vision encoder, a multimodal projector and a backbone LLM, of which our work is based on. The training paradigm for LLaVA-like VLMs consists of two portions, pretraining and supervised finetuning. In the pretraining stage, all but the multimodal projector is frozen, where the projector is then trained on image-text pairs to learn the mapping from visual to token space. Then, in the supervised finetuning stage, only the vision encoder is frozen, while both the projector and backbone LLM are tuned, extending the LLM to multi-modal tasks.
>
> **Training procedure of EMMA.** Due to the effectiveness of the Mamba LLM, Cobra discovered that Mamba-based MLLMs do not benefit much from the pretraining stage. Thus, we follow the same procedure of Cobra and discard the pretraining stage altogether, and only conduct supervised finetuning for two epochs. For the additional structures introduced, we simply add the structural and hierarchical modules to our model, and jointly optimize with the original text generation objective. This simplifies the training of our methods, where only finetuning data is used to end-to-end finetune our entire model. We also do not require a separate stage nor additional data for the training of decoder or MFF module, unlike transformer-based EMU [15] and EMU2 [16].

---

> ### Author Response · Authors · 2024-11-21
> **Q4. Furthermore, when comparing EMMA to other methods, the additional training costs should be listed separately to ensure a fair comparison**
>
> **Q4. Furthermore, when comparing EMMA to other methods, the additional training costs should be listed separately to ensure a fair comparison**
>
> **Fair comparison of training costs.** We have provided the complete breakdown of training and testing costs in the table for Q1. The visual encoder has the same train-time and evaluation-time values due to it being frozen and un-optimized during either training and testing. All other components utilize less memory and time resources in evaluation, compared to training. The newly introduced structures, decoder and MFF, are discarded during testing. Note that the training time for decoder is relatively large compared to other structures given its small size. This is due to the calculation of respective gradients that concern the update of LLM parameters due to pixel-alignment loss.

---

> ### Author Response · Authors · 2024-11-21
> **Q5. Since EMMA alters the original MLLM architecture, its inference complexity will also increase.**
>
> **Q5. Since EMMA alters the original MLLM architecture, its inference complexity will also increase. Table 4 should include a breakdown of the increased complexity due to hierarchical alignment. For fairness, Table 4 should also provide EMMA’s inference speed when using Mamba LLM-2.8B.**
>
> **Alterations to inference speed.** Because both the structural alignment and hierarchical alignment consider visual features, which are used to boost the quality of visual latents and encourage better multi-modal alignment during the *training* stage, they are discarded during inference as they are not required for visual question answering. Thus, as shown in the provided component analysis table, they do not introduce additional complexity during inference.
>
> **EMMA-V1 Inference Time.** As shown in the table for Q1, during inference, the decoder is discarded, and only the original LLM outputs are considered. Thus, the reason why EMMA-V1’s inference speed is not reported is due to the fact that both EMMA-V1 and Cobra uses the Mamba LLM-2.8B backbone, so they should have the same inference speeds. We will include the EMMA-V1 inference time in the final manuscript.

---

> ### Author Response · Authors · 2024-11-21
> **Q6. Provide more detailed analysis of how their method addresses specific challenges in multimodal learning that previous methods have struggled with.**
>
> **Q6. EMMA’s approach is somewhat simplistic, as it merely enhances the consistency between deep and shallow features and the original image at the output stage. This method requires extra training, substantial computational resources, and introduces additional structures during deployment. As a result, the technical contribution appears somewhat trivial. It would be better if this paper could provide more detailed analysis of how their method addresses specific challenges in multimodal learning that previous methods have struggled with.**
>
> **Technical Contributions.** We believe our main contribution of this paper lies in achieving better alignment of visual and textual features, specifically in large-scaled multi-modal Mamba models. In the paper, we first observe the inconsistent performance of Mamba models with respect to visual and textual inputs when scaled-up in parameters. In specific, Mamba models tend to experience a significant performance decline in processing visual data when scaled-up, despite them being able to seamlessly handle textual data in large-scaled settings. We analyze this issue in two perspectives, both architecturally and empirically.
>
> **Architectural imbalance.** The inherent linear scanning mechanism of Mamba poses challenges in effectively extracting visual features on the same level as textual features. On one hand, transformer-based [17] approaches utilize the self-attention mechanism that extracts global relationships between different parts of the input sequence. They also utilize a position encoding for learning spatial relationships of visual tokens. On the other hand, CNN-based approaches [18] utilize local receptive fields to connect each neuron to a local region of the input data, allowing interaction of neighboring pixels in all directions. The effectiveness of both methods in processing visual information lies in their ability to account for global, or to the least, local clusters of the visual input. However, Mamba utilizes a selective-scan mechanism that only processes sequences in a linear order. While Mamba models achieve excellent performances in textual data, effectively capturing long-term dependencies in multiple language benchmarks [19], Mamba models possess difficulties in handling image data, where pixel-wise relationships may span multiple rows and columns. Visual Mamba models have been proposed to counteract this issue through combining additional scans from different directions [20] and utilizing a similar position encoding as ViTs [21]. However, they nevertheless still utilize the linear selective-scan mechanism. In the case where features possess large embedding dimensions, which is typical in MLLM scenarios, this becomes hard for the model to extract visual cues that are far apart spatially, as the degree of freedom between each visual token across different rows and columns become huge.
>
>
>
> **Empirical imbalance.** Mamba models experiences an imbalance in the empirical evaluation of vision tasks and textual tasks, when scaled up in parameters. [22] observes the performance degradation of visual Mamba models when scaled up. Below is a table taken directly from their paper.
>
>
>
> | Model        | VIM-B | VIM-L | VIM-H |
> |--------------|-------|-------|-------|
> | Size (M)     | 98    | 340   | 755   |
> | ImageNet Acc.| 81.2  | 81.0  | Collapsed |
>
> As shown, plain Mamba models experience a severe degradation in processing visual features when scaled up. In the 700M parameter-scale, the visual Mamba collapses entirely, rendering the need for effective approaches to alleviate this performance degradation, specifically in Mamba-based models. On the other hand, plain Mamba models do not experience performance degradation in processing textual data, as shown from the experiment results in the original Mamba papers [5, 6].
>
> | Model        | Mamba-790M | Mamba-1.4B | Mamba-2.8B | Mamba2-780M | Mamba2-1.3B | Mamba2-2.7B |
> |--------------|------------|------------|------------|-------------|-------------|-------------|
> | Size (M)     | 790        | 1,400      | 2,800      | 780         | 1,300       | 2,700       |
> | HellaSwag    | 55.1       | 59.1       | 66.1       | 54.9        | 59.9        | 66.6        |
> | LAMBADA      | 62.7       | 64.9       | 69.2       | 61.7        | 65.7        | 69.7        |
>
>
> Thus, we observe an imbalance of the processing capabilities of Mamba on visual and textual inputs, when scaled up in billion-parameter models. This phenomenon renders the need for more effective visual processing in large-scaled Mamba models to ensure that visual and textual features are adequately aligned.

---

> ### Author Response · Authors · 2024-11-21
> **Q6. Continued**
>
> **Application of Mamba in MLLMs.** On the other hand, to deploy Mamba models in MLLM settings, where large-scaled Mamba models are necessary in processing both textual and visual information, it is crucial to preserve a balance in the quality of visual and textual features to achieve multi-modal alignment. Previous Mamba MLLM works such as VL Mamba and Cobra directly replaces the transformer LLM with the Mamba LLM model without addressing this imbalance of feature quality. Still, the performance of these models show the potential of extending mambas to the realm of MLLMs, especially with their fast inference speeds.
>
>
>
> **Ineffectiveness of transformer-based techniques.** Furthermore, as a preliminary study for this work, we have also tried transformer-based multimodal alignment approaches that enhance visual features or encourage multimodal alignment, such as visual-feature compression [23], masked-alignment [24, 25], and contrastive losses [26, 27], all of which fail in the Mamba LLM setting. Here, we show the respective model performances on the TextVQA dataset, which we find is a good indicator for overall model performance.
>
> | Model        | EMMA-V1 | EMMA-V2 | Cobra | Compression | Masked | Contrastive |
> |--------------|---------|---------|-------|-------------|--------|-------------|
> | TextVQA      | 57.2    | 56.2    | 52.4  | 45.8        | 46.1   | 50.2        |
>
> As shown, transformer-based approaches tailored for multimodal alignment in MLLM settings may in fact negatively impact model performance, which renders the necessity to devise Mamba-specific multi-modal alignment approaches.
>
>
> **Novelty of our approach.** On the other hand, autoregressive pretraining on the visual features are shown to improve visual Mamba models [22], and for the first time allows visual Mamba models to scale up to large-sized models (600M). Given the effectiveness of autoregressive pretraining in promoting the single-modal visual Mamba model, we extend this approach to the multimodal LLM setting. Subsequently, we propose the pixel-wise alignment loss to encourage the generated autoregressive features to structurally align with the input image. Furthermore, when visualizing the LLM features, we notice a gradual decline of the intermediate visual features, causing the final generated visual feature to degrade. In response, we propose the multi-scale feature fusion module to tackle this issue, leading to better preservation of intermediate visual features, inducing better multimodal alignment of visual and textual features. In essence, our method specifically tackles the issue of multi-modal alignment in Mamba models, while the direct application of transformer-based multimodal methods have failed. As a result, EMMA demonstrates improved performances on a variety of metrics, and most importantly, significantly reduces the degree of visual hallucinations in the HallusionBench benchmark, surpassing transformer models with twice the size. We believe this study provides groundwork for further research of Mamba models in the MLLM framework as a potential replacement for transformer-based approaches, with their comparable performance and superior inference latency.
>
>
> **Potential impact and application of our observations and techniques.** On one hand, we believe the insights introduced in this paper will help inspire future research of mamba models in the realm of MLLM and other downstream tasks. Specifically, our approach to strengthen the quality of visual features in large-scaled mamba models may provide future researchers with a framework for multimodal alignment in Mamba models. Our light-weight Mamba-based decoder also serves as an efficient means to conduct feature reconstruction in the visual modality. Furthermore, our multi-scale feature fusion module can be an inspiration for combining same-modality features in future Mamba-based frameworks. On the other hand, we aim for our research to pitch Mamba as a potential contender among transformer-based architectures. Given Mamba's swift inference capabilities, we show in this work that Mamba-based MLLMs can achieve competitive performance with transformer models, and underscores its potential for widespread adoption and integration into diverse AI applications.

---

> ### Author Response · Authors · 2024-11-21
> **References**
>
> 1. Hezheng Lin, Xing Cheng, Xiangyu Wu, and Dong Shen. Cat: Cross attention in vision transformer. In 2022 IEEE international conference on multimedia and expo (ICME), pp. 1–6. IEEE, 2022.
> 2. Jean-Baptiste Alayrac, Jeff Donahue, Pauline Luc, Antoine Miech, Iain Barr, Yana Hasson, Karel Lenc, Arthur Mensch, Katherine Millican, Malcolm Reynolds, et al. Flamingo: a visual language model for few-shot learning. Advances in neural information processing systems, 35:23716–23736, 2022.
> 3. Kaibing Chen, Dong Shen, Hanwen Zhong, Huasong Zhong, Kui Xia, Di Xu, Wei Yuan, Yifei Hu, Bin Wen, Tianke Zhang, et al. Evlm: An efficient vision-language model for visual understanding. arXiv preprint arXiv:2407.14177, 2024.
> 4. Zonghao Guo, Ruyi Xu, Yuan Yao, Junbo Cui, Zanlin Ni, Chunjiang Ge, Tat-Seng Chua, Zhiyuan Liu, and Gao Huang. Llava-uhd: an lmm perceiving any aspect ratio and high-resolution images. In European Conference on Computer Vision, pp. 390–406. Springer, 2025.
> 5. Albert Gu and Tri Dao. Mamba: Linear-time sequence modeling with selective state spaces. arXiv preprint arXiv:2312.00752, 2023.
> 6. Tri Dao and Albert Gu. Transformers are ssms: Generalized models and efficient algorithms through structured state space duality. arXiv preprint arXiv:2405.21060, 2024.
> 7. Opher Lieber, Barak Lenz, Hofit Bata, Gal Cohen, Jhonathan Osin, Itay Dalmedigos, Erez Safahi, Shaked Meirom, Yonatan Belinkov, Shai Shalev-Shwartz, et al. Jamba: A hybrid transformer-mamba language model. arXiv preprint arXiv:2403.19887, 2024.
> 8. Jamba Team, Barak Lenz, Alan Arazi, Amir Bergman, Avshalom Manevich, Barak Peleg, Ben Aviram, Chen Almagor, Clara Fridman, Dan Padnos, et al. Jamba-1.5: Hybrid transformer-mamba models at scale. arXiv preprint arXiv:2408.12570, 2024.
> 9. Paolo Glorioso, Quentin Anthony, Yury Tokpanov, James Whittington, Jonathan Pilault, Adam Ibrahim, and Beren Millidge. Zamba: A compact 7b ssm hybrid model. arXiv preprint arXiv:2405.16712, 2024.
> 10. Soham De, Samuel L Smith, Anushan Fernando, Aleksandar Botev, George Cristian-Muraru, Albert Gu, Ruba Haroun, Leonard Berrada, Yutian Chen, Srivatsan Srinivasan, et al. Griffin: Mixing gated linear recurrences with local attention for efficient language models. arXiv preprint arXiv:2402.19427, 2024.
> 11. Lucidrains. Linear attention transformer. 2021. URL https://github.com/lucidrains/linear-attention-transformer.
> 12. Sinong Wang, Belinda Z Li, Madian Khabsa, Han Fang, and Hao Ma. Linformer: Self-attention with linear complexity. arXiv preprint arXiv:2006.04768, 2020.
> 13. A. Katharopoulos, A. Vyas, N. Pappas, and F. Fleuret. Transformers are rnns: Fast autoregressive transformers with linear attention. In Proceedings of the International Conference on Machine Learning (ICML), 2020. URL https://arxiv.org/abs/2006.16236.
> 14. Haotian Liu, Chunyuan Li, Qingyang Wu, and Yong Jae Lee. Visual instruction tuning. Advances in neural information processing systems, 36, 2024.
> 15. Quan Sun, Qiying Yu, Yufeng Cui, Fan Zhang, Xiaosong Zhang, Yueze Wang, Hongcheng Gao, Jingjing Liu, Tiejun Huang, and Xinlong Wang. Emu: Generative pretraining in multimodality. In The Twelfth International Conference on Learning Representations, 2023a.
> 16. Quan Sun, Qiying Yu, Yufeng Cui, Fan Zhang, Xiaosong Zhang, Yueze Wang, Hongcheng Gao, Jingjing Liu, Tiejun Huang, and Xinlong Wang. Generative pretraining in multimodality. arXiv preprint arXiv:2307.05222, 2023b.
> 17. Alexey Dosovitskiy. An image is worth 16x16 words: Transformers for image recognition at scale. arXiv preprint arXiv:2010.11929, 2020.
> 18. Alex Krizhevsky, Ilya Sutskever, and Geoffrey E Hinton. Imagenet classification with deep convolutional neural networks. Advances in neural information processing systems, 25, 2012.
> 19. Yi Tay, Mostafa Dehghani, Samira Abnar, Yikang Shen, Dara Bahri, Philip Pham, Jinfeng Rao, Liu Yang, Sebastian Ruder, and Donald Metzler. Long range arena: A benchmark for efficient transformers. arXiv preprint arXiv:2011.04006, 2020.
> 20. Yue Liu, Yunjie Tian, Yuzhong Zhao, Hongtian Yu, Lingxi Xie, Yaowei Wang, Qixiang Ye, and Yunfan Liu. Vmamba: Visual state space model. arXiv preprint arXiv:2401.10166, 2024.
> 21. Lianghui Zhu, Bencheng Liao, Qian Zhang, Xinlong Wang, Wenyu Liu, and Xinggang Wang. Vision mamba: Efficient visual representation learning with bidirectional state space model. arXiv preprint arXiv:2401.09417, 2024.
> 22. Sucheng Ren, Xianhang Li, Haoqin Tu, Feng Wang, Fangxun Shu, Lei Zhang, Jieru Mei, Linjie Yang, Peng Wang, Heng Wang, et al. Autoregressive pretraining with mamba in vision. arXiv preprint arXiv:2406.07537, 2024.
> 23. Wentong Li, Yuqian Yuan, Jian Liu, Dongqi Tang, Song Wang, Jie Qin, Jianke Zhu, and Lei Zhang. Tokenpacker: Efficient visual projector for multimodal llm. arXiv preprint arXiv:2407.02392, 2024.

---

> ### Author Response · Authors · 2024-11-21
> **References, continued.**
>
> 24. David Mizrahi, Roman Bachmann, Oguzhan Kar, Teresa Yeo, Mingfei Gao, Afshin Dehghan, and Amir Zamir. 4m: Massively multimodal masked modeling. Advances in Neural Information Processing Systems, 36, 2024.
> 25. Yonatan Bitton, Gabriel Stanovsky, Michael Elhadad, and Roy Schwartz. Data efficient masked language modeling for vision and language. arXiv preprint arXiv:2109.02040, 2021.
> 26. Sheng Shen, Liunian Harold Li, Hao Tan, Mohit Bansal, Anna Rohrbach, Kai-Wei Chang, Zhewei Yao, and Kurt Keutzer. How much can clip benefit vision-and-language tasks? arXiv preprint arXiv:2107.06383, 2021.
> 27. Alec Radford, Jong Wook Kim, Chris Hallacy, Aditya Ramesh, Gabriel Goh, Sandhini Agarwal, Girish Sastry, Amanda Askell, Pamela Mishkin, Jack Clark, et al. Learning transferable visual models from natural language supervision. In International conference on machine learning, pp.8748–8763. PMLR, 2021

---

> ### Author Response · Authors · 2024-11-24
> **Looking forward to further discussions!**
>
> Dear Reviewer yCEA,
>
> Thank you for your insightful comments. We were wondering if our response and revision have resolved your concerns. We have attempted to address your initial questions through our replies and are eager to clarify any further points you might raise. Please feel free to provide additional feedback. We greatly appreciate your continued engagement.
>
> Best regards,
> Authors

---

> ### Comment · Area_Chair_5dM9 · 2024-11-26
>
> Dear Reviewer yCEA,
>
> Could you kindly review the rebuttal thoroughly and let us know whether the authors have adequately addressed the issues raised or if you have any further questions.
>
> Best,
>
> AC

---

### Author Response · Authors · 2024-11-14
**A grateful thanks to all reviewers.**

We express our gratitude to all the reviewers for their meticulous reviews and valuable feedback on our paper. The effort put forth in posing insightful questions will greatly aid us in refining and clarifying our work. We are pleased to learn that the motivation has been effectively clarified (as noted by Reviewer T8Kc and yy1F), the paper is well-written (highlighted by Reviewer T8Kc and yy1F), and the experiments substantiate the efficacy of our approach (acknowledged by all reviewers). Over the next few days, we will diligently work towards addressing all raised concerns in order to enhance the quality and comprehensiveness of this work. Thanks again for reviewing our work!

---

### Author Response · Authors · 2024-11-25
**A huge thanks to all reviewers, we look forward to hearing from you.**

We thank the reviewers again for their valuable feedback. We are grateful that most reviewers are quite affirmative of our overall contributions, including the novelty and motivation ("Pixel-wise alignment loss is interesting" by 5nFQ, "Motivation Well Clarified” by T8Kc, "The imbalance ... does make sense to investigate ..." by yy1F), the extensiveness of empirical evaluation ("The paper presents comprehensive experiments" by T8Kc, "The experiments are sufficient" by yy1F), the effectiveness of the proposed method ("simple yet effective" by yCEA, "experimental results seem promising" by 5nFQ, "robustness of the proposed approach" by T8Kc, "effectiveness of the proposed method" by yy1F), and clarity and soundness of the paper ("Well Written" by T8Kc, "well-written and easy to follow" by yy1F).

We summarize the main contributions of our paper:
1. We observe and discover the difference in performance of mamba-based models in vision vs. language tasks, especially in large-scale settings. This renders the design of mamba-specific architectures in order to bring out the full potential of mamba models in MLLM tasks.
2. We are the first to extend from the vanilla MLLM framework for mamba models and propose mamba-specific modules to address the imbalance between visual and textual latents, enabling better multi-modal fusion of large-scaled mamba-based models.
3. Due to hierarchical and structural alignment, EMMA is able to achieve better performance than current Mamba-based MLLMs and competitive performance with transformer-based approaches with similar scales. Furthermore, our model demonstrates superior capabilities in reducing hallucinations, where it outperforms all current models <4B in POPE in the OpenCompass VLM leaderboard and outperforms all current models <10B in HallusionBench in the OpenCompass VLM leaderboard. In terms of inference speed, our model is 3 times faster than current similar-sized transformer models. This presents Mamba as a promising contender in the realm of efficient general AI models that require fast response speeds.

We have also made a number of comments to address all reviewers' suggestions and concerns. A short summary of the
replies are made as:

1. Computation analysis: We provide a comparison of each module in terms of computational cost during training and inference. Our newly proposed structural and hierarchical alignment only occurs during training and is discarded during the inference stage.
2. Training stage clarification: Our method only includes a supervised fine-tuning stage that tunes all LLM, MLP projector, and decoder parameters in an end-to-end fashion. A summary of the preprocessing of training data is also included in the revised paper.
3. Motivation clarification: We analyze mamba models both architecturally and empirically to demonstrate the imbalance in the quality of visual and textual latents. This phenomenon renders the need for more effective visual processing in large-scale Mamba models to ensure that visual and textual features are adequately aligned.
4. Additional visualizations: We include additional visualizations to further demonstrate the effectiveness of our method.

Since the rebuttal deadline is incoming, please let us know if our replies have addressed your concerns. We are more than delighted to have further discussions and improve our manuscript. If our response has addressed your concerns, we would be grateful if you could re-evaluate our work.

---

### Meta-Review · Area_Chair_5dM9 · 2024-12-18

**Metareview:**

The paper introduces Empowering Multi-modal Mamba with Structural and Hierarchical Alignment (EMMA), enhancing Mamba multi-modal large language models (MLLM) by improving their ability to extract fine-grained visual information. This paper is well written and easy to understand. Most of reviewers point out that the pixel-wise alignment loss is interesting and useful, which is demonstrated by doing extensive experiments and showing good performance. The authors are required to update their final version of papers considering the reviews. All reviewers agree to accept this paper, thus I lean to make the recommondation of acceptance.

**Additional Comments On Reviewer Discussion:**

Many concerns are raised and it seems like these concerns are well addressed during the period of rebuttal.

---

### Decision · Program_Chairs · 2025-01-22

Accept (Poster)